# ACMTF-R: Supervised multi-omics data integration uncovering shared and distinct outcome-associated variation

**Geert Roelof van der Ploeg**[1]*, **Fred T. G. White**[1], **Rasmus Riemer Jakobsen**[2], **Johan A. Westerhuis**[1], **Anna Heintz-Buschart**[1], **Age K. Smilde**[1]

**1** Biosystems Data Analysis, Swammerdam Institute for Life Sciences, University of Amsterdam, Amsterdam, The Netherlands, **2** Department of Food Science, University of Copenhagen, Rolighedsvej, Frederiksberg C, Denmark

* roel@simula.no

## Abstract

The rapid growth of high-dimensional biological data has necessitated advanced data fusion techniques to integrate and interpret complex multi-omics and longitudinal datasets. Shared and unshared structure across such datasets can be identified in an unsupervised manner with Advanced Coupled Matrix and Tensor Factorization (ACMTF), but this cannot be related to an outcome. Conversely, N-way Partial Least Squares (NPLS) is supervised and captures outcome-associated variation but cannot identify shared and unshared structure. To bridge the gap between data exploration and prediction, we introduce ACMTF-Regression (ACMTF-R), an extension of ACMTF that incorporates a regression step, allowing for the simultaneous decomposition of multi-way data while explicitly capturing variation associated with a dependent variable. We present a detailed mathematical formulation of ACMTF-R, including its optimisation algorithm and implementation. Through extensive simulations, we systematically evaluate its ability to recover a small **y**-related component shared between multiple blocks, its robustness to noise, and the impact of the tuning parameter ($\pi$) which controls the balance between data exploration and outcome prediction. Our results demonstrate that ACMTF-R can robustly identify the **y**-related component, correctly identifying outcome-associated shared and distinct variation, distinguishing it from existing approaches such as NPLS and ACMTF. The development of ACMTF-R was motivated by a real-world dataset investigating how maternal pre-pregnancy BMI affects the human milk microbiome, human milk metabolome, and infant faecal microbiome. Emerging evidence suggests that inter-generational transfer of maternal obesity may affect multiple omics layers, highlighting the need to identify outcome-associated variation. The applicability of ACMTF-R is therefore validated by applying it to this multi-omics dataset. ACMTF-R successfully identifies novel mother-infant relationships associated with maternal pre-pregnancy BMI, underscoring its utility in multi-omics research. Our findings establish ACMTF-R as

**Data availability statement:** The CMTFtoolbox R package is available on CRAN, with a development version on GitHub (https://doi.org/10.5281/zenodo.16633119). The underlying code for this paper is supplied in a GitHub repository at https://doi.org/10.5281/zenodo.16633167.

**Funding:** GRvdP was funded by a grant from the University of Amsterdam, Research Priority Area on Personal Microbiome Health. FW was funded by a grant from the University of Amsterdam, Data Science Centre.

**Competing interests:** The authors have declared that no competing interests exist.

a versatile tool for multi-way data fusion, offering new insights into complex biological systems by integrating common, local, and distinct variation in the context of a dependent variable.

## Introduction

The advent of high-throughput omics technologies has led to a rapid growth of complex, high-dimensional biological datasets, including genomics [1–3], transcriptomics [4], proteomics [5], metabolomics [4], and microbiome [6]. When measured on the same samples, these multi-omics datasets give researchers the opportunity to obtain a systems-level understanding of biological processes [5]. However, their integration and analysis pose significant challenges due to data heterogeneity [5,7], high dimensionality [4,6], and complex interactions between different omics datasets [8,9]. Advanced data analytical methods are required to extract meaningful biological insights by distinguishing variation that is shared across datasets from variation that is dataset-specific [10–13].

A critical need in the analysis of multi-omics datasets is the identification of common, local and distinct sources of variation (**Fig 1**; [8–10,14]). Common variation is shared across all datasets, local variation is shared between a subset of datasets, and distinct variation is unique to a dataset. Identifying the Common, Local and Distinct (CLD) structure of a multi-omics dataset elucidates the complex biological processes that occur within the system.

Advanced Coupled Matrix and Tensor Factorization (ACMTF) has emerged as a powerful multi-way data integration approach capable of uncovering common, local, and distinct sources of variation across datasets [10,15–18]. However, ACMTF is unsupervised and therefore does not allow for the identification of shared and distinct variation related to a dependent variable. N-way Partial Least Squares (NPLS) [19] is a supervised method that identifies variation related to a dependent variable, but is limited to analysing one dataset at a time and thus cannot simultaneously identify common, local, and distinct variation across multiple datasets.

This work introduces Advanced Coupled Matrix and Tensor Factorization Regression (ACMTF-R), building on four strands of prior work: (i) the separation of joint and individual structures in matrices [14,20] and tensors [10], (ii) relating latent variables to an outcome [19,21], (iii) performing a coupled decomposition for any combination of matrices and tensors [15,22,23], and (iv) a tuning parameter ($\pi$) that balances data reconstruction and prediction [24–26]. This approach enables unified extraction of common, local, and distinct variation across multiple data blocks while modelling an outcome-associated structure.

In this study, we first provide a detailed overview of the ACMTF framework, and the methodological development represented by ACMTF-R. We then present a simulation-based approach to systematically evaluate the ability of ACMTF-R to capture a small, hidden, *y*-related component under different noise conditions and tuning parameter values. Finally, we apply ACMTF-R to a real-world dataset, integrating human milk (HM) microbiome, HM metabolome, and infant gut microbiome data

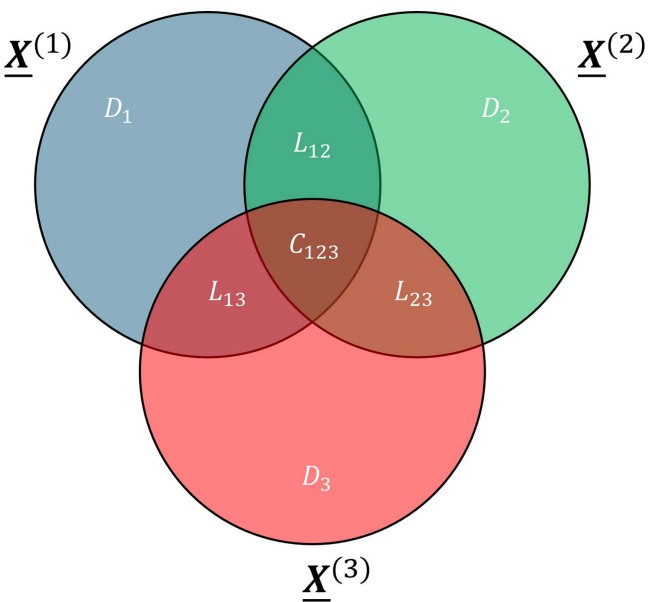

**Fig 1. Common, local and distinct variation.** Common variation $C_{123}$ is shared between all datasets $\underline{\mathbf{X}}^{(1)}$, $\underline{\mathbf{X}}^{(2)}$ and $\underline{\mathbf{X}}^{(3)}$. Local variation is shared between some of the datasets. Hence $\mathbf{L}_{12}$ is shared between $\underline{\mathbf{X}}^{(1)}$ and $\underline{\mathbf{X}}^{(2)}$, but not $\underline{\mathbf{X}}^{(3)}$. Similarly, $\mathbf{L}_{13}$ is shared between $\underline{\mathbf{X}}^{(1)}$ and $\underline{\mathbf{X}}^{(3)}$, but not $\underline{\mathbf{X}}^{(2)}$, and $\mathbf{L}_{23}$ is shared between $\underline{\mathbf{X}}^{(2)}$ and $\underline{\mathbf{X}}^{(3)}$, but not $\underline{\mathbf{X}}^{(1)}$. Distinct variation is unique to one dataset: $\mathbf{D}_1$ for block $\underline{\mathbf{X}}^{(1)}$, $\mathbf{D}_2$ for block $\underline{\mathbf{X}}^{(2)}$, and $\mathbf{D}_3$ for block $\underline{\mathbf{X}}^{(3)}$. Together, they make up the Common, Local and Distinct (CLD) structure of the data.

to study how maternal pre-pregnancy body mass index (ppBMI) is associated with these data. Our findings highlight the potential of ACMTF-R as a versatile tool for multi-omics data integration, facilitating the discovery of biologically meaningful common, local and distinct variation, while accommodating both exploratory and predictive research objectives.

## Methods

### Notation and definitions

We briefly define the mathematical notation that will be used throughout this paper. The notation proposed by Kiers [27] is followed with some minor extensions for multi-omics. Scalars are denoted by lower-case letters (e.g., $a$, $b$, $c$). Column vectors are denoted by boldface lowercase letters (e.g., $\mathbf{a}$, $\mathbf{b}$, $\mathbf{c}$). Matrices are denoted by boldface capital letters (e.g., $\mathbf{X}$, $\mathbf{Y}$, $\mathbf{Z}$). Three-way arrays are denoted by underlined boldface capital letters (e.g., $\underline{\mathbf{X}}$, $\underline{\mathbf{Y}}$, $\underline{\mathbf{Z}}$). While $\underline{\mathbf{X}}$ can more generally be used to indicate four-way or higher-order arrays, this paper does not discuss such data.

The characters $I$, $J$, and $K$ are reserved to indicate the first, second and third mode of an array. A two-way array $\mathbf{X}$ will be assumed to be of size $I \times J$, while a three-way array $\underline{\mathbf{X}}$ will be assumed to be of size $I \times J \times K$. Similarly, lowercase characters $i$, $j$, and $k$ will be used as indices for the first, second, and third modes, respectively. Columns of a matrix are vectors with a subscript of their index (e.g., $\mathbf{x}_1$, $\mathbf{x}_2$, ...,$\mathbf{x}_J$ for some matrix $\mathbf{X}$). Similarly, the i-th row and j-th column entry is denoted as a scalar with subscripts of their indices (e.g., $x_{ij}$). For three-way arrays, the entries per mode are given as subscripts (e.g., $x_{ijk}$).

The index $p = 1, \ldots, P$ is used to indicate the block number, which is denoted by a superscript (e.g., $\underline{\mathbf{X}}^{(1)}$, $\underline{\mathbf{X}}^{(2)}$, ..., $\underline{\mathbf{X}}^{(P)}$) and the size is indicated likewise (e.g., the size of $\underline{\mathbf{X}}^{(p)}$ is $I \times J^{(p)} \times K^{(p)}$). In ACMTF and ACMTF-R $\underline{\mathbf{X}}^{(1)}$, $\underline{\mathbf{X}}^{(2)}$, ..., $\underline{\mathbf{X}}^{(P)}$ can be any mixture of two-way or three-way arrays, but this study focuses only on the case where all blocks are three-way arrays with a shared subject mode. Hence the superscript $(p)$ is not used to describe the size of the first mode $I$.

Mathematically it is convenient to matricise a three-way array $\underline{\boldsymbol{X}}$, as this transformation facilitates the description of multi-way models using matrix notation [27–29]. There are three different variants of matricisation, depending on which mode of the three-way array is preserved as rows in the resulting matrix. Hence, first mode matricisation yields a matrix $\boldsymbol{X}_a$ of size $I \times JK$. Similarly, second mode matricization produces a matrix $\boldsymbol{X}_b$ of size $J \times IK$, while third mode matricization generates a matrix $\boldsymbol{X}_c$ of size $K \times IJ$.

The vectorisation of an $I \times J$ matrix $\boldsymbol{X}$ is denoted $\text{vec}(\boldsymbol{X})$, such that $\text{vec}(\boldsymbol{X}) = \left[\boldsymbol{x}_1^T\, \boldsymbol{x}_2^T \ldots \boldsymbol{x}_J^T\right]^T$. The Hadamard product [30] of two equally sized arrays $\boldsymbol{X}$ and $\boldsymbol{Y}$ is denoted $\boldsymbol{X} * \boldsymbol{Y}$, such that $(\boldsymbol{X} * \boldsymbol{Y})_{ij} = x_{ij}y_{ij}$. The Kronecker product, denoted $\otimes$, of two matrices $\boldsymbol{X}$ with size $I \times J$ and $\boldsymbol{Y}$ with size $K \times L$ produces a block matrix containing all pairwise products of their entries following $(\boldsymbol{X} \otimes \boldsymbol{Y})_{ik,jl} = x_{ij}y_{kl}$. Thus, we have:

$$\boldsymbol{X} \otimes \boldsymbol{Y} = \begin{bmatrix} x_{11}\boldsymbol{Y} & \cdots & x_{1J}\boldsymbol{Y} \\ \vdots & \ddots & \vdots \\ x_{I1}\boldsymbol{Y} & \cdots & x_{IJ}\boldsymbol{Y} \end{bmatrix}$$

(1)

where the outcome of $\boldsymbol{X} \otimes \boldsymbol{Y}$ is size $IK \times JL$.

The Khatri-Rao product, denoted $\odot$, of two matrices $\boldsymbol{X}$ of size $I \times R$ and $\boldsymbol{Y}$ of size $J \times R$ is defined as the column-wise Kronecker product [31,32]

$$\boldsymbol{X} \odot \boldsymbol{Y} = \begin{bmatrix} | & | & \cdots & | \\ \boldsymbol{x}_1 \otimes \boldsymbol{y}_1 & \boldsymbol{x}_2 \otimes \boldsymbol{y}_2 & \cdots & \boldsymbol{x}_R \otimes \boldsymbol{y}_R \\ | & | & \cdots & | \end{bmatrix} = \begin{bmatrix} | & | & \cdots & | \\ \text{vec}(\boldsymbol{y}_1\boldsymbol{x}_1^T) & \text{vec}(\boldsymbol{y}_2\boldsymbol{x}_2^T) & \cdots & \text{vec}(\boldsymbol{y}_R\boldsymbol{x}_R^T) \\ | & | & \cdots & | \end{bmatrix}$$

where the outcome of $\boldsymbol{X} \odot \boldsymbol{Y}$ has size $IJ \times R$.

The Frobenius (or Euclidean) norm of a three-way array $\boldsymbol{X}$ is defined as:

$$\|\underline{\boldsymbol{X}}\| = \sqrt{\sum_{i=1}^{I}\sum_{j=1}^{J}\sum_{k=1}^{K} x_{ijk}^2}$$

(2)

and the notation $\| \cdot \|$ will also be used to refer to the Frobenius norm for matrices and vectors, respectively [28]. The Moore-Penrose inverse [33] of $\boldsymbol{X}$ is denoted $\boldsymbol{X}^{+}$.

In this paper we follow the canonical polyadic form for defining multi-way arrays as the sum of rank-one arrays with normalized factor matrices and a scalar $\lambda$ to capture the magnitude per term [15,34–36]. The notation $\left[\!\left[\lambda_p\,;\boldsymbol{A},\,\boldsymbol{B}^{(p)},\,\boldsymbol{C}^{(p)}\right]\!\right]$ is used to describe the model of the three-way array $\underline{\boldsymbol{X}}^{(p)}$, defined elementwise as

$$x_{ijk}^{(p)} = \sum_{f=1}^{F} \lambda_{pf} a_{if} b_{jf}^{(p)} c_{kf}^{(p)} + e_{ijk}^{(p)}$$

(3)

where columns of the loading matrices $\boldsymbol{A}, \boldsymbol{B}^{(p)}, \boldsymbol{C}^{(p)}$ are norm 1 and the column vector $\lambda_p$ is used to modify the size of each identified component $f = 1, \ldots, F$ to norm $\lambda_{pf}$. For a two-way array, equation (3) reduces to $[\![\boldsymbol{A}, \boldsymbol{B}]\!] = \boldsymbol{A}\boldsymbol{B}^T$, assuming that $\lambda = 1$ with size $1 \times F$ and that the singular values are multiplied into $\boldsymbol{A}$ to stay in line with common practice.

## Advanced Coupled Matrix and Tensor Factorization (ACMTF)

Advanced Coupled Matrix and Tensor Factorization is an all-at-once joint factorisation approach that is applicable to mixtures of matrices and tensors [10,15,16,37]. The method seeks a decomposition of the datasets $\underline{\boldsymbol{X}}^{(1)}, \underline{\boldsymbol{X}}^{(2)}, \ldots, \underline{\boldsymbol{X}}^{(P)}$ as $\hat{\underline{\boldsymbol{X}}}^{(p)} = \left[\lambda_p; \boldsymbol{A}, \boldsymbol{B}^{(p)}, \boldsymbol{C}^{(p)}\right]$ such that loading matrices corresponding to shared modes are equal between the blocks. In this

paper, we will assume that only the first mode is shared between all blocks. In that case, the matrix containing the first mode loadings $A$ will be the same for every block $p$ and the superscript can be left out.

The loss function for ACMTF is defined as

$$f\left(\alpha,\beta,\varepsilon, \Lambda, A, B^{(1)}, C^{(1)}, \ldots, B^{(P)}, C^{(P)}\right) = \sum_{p=1}^{P} \left\| \underline{X}^{(p)} - \hat{\underline{X}}^{(p)} \right\|^2$$
$$+ \beta \sum_{p=1}^{P} \sum_{f=1}^{F} \sqrt{\lambda_{pf}^2 + \epsilon}$$
$$+ \alpha \sum_{p=1}^{P} \sum_{f=1}^{F} \left( \|a_f\| - 1 \right)^2$$
$$+ \alpha \sum_{p=1}^{P} \sum_{f=1}^{F} \left( \left\| b_f^{(p)} \right\| - 1 \right)^2$$
$$+ \alpha \sum_{p=1}^{P} \sum_{f=1}^{F} \left( \left\| c_f^{(p)} \right\| - 1 \right)^2$$

(4)

where the $P \times F$ matrix $\Lambda = [\lambda_1 \ \lambda_2 \ldots \lambda_P]^T$ scales the factors per component and per data block, $\alpha > 0$ is the penalty setting for norm-1 components, $\beta > 0$ is the sparsity penalty setting for $\Lambda$, $F$ is the number of components, and $p$ is the block index. Here the 1-norm loss term for the elements in $\Lambda$ is replaced with a differentiable approximation $\|\lambda_p\|_1 \approx \sum_{f=1}^{F} \sqrt{\lambda_{pf}^2 + \epsilon}$ for a sufficiently small $\epsilon > 0$ [37,38]. The originally suggested default settings are $\alpha = 1$, $\beta = 10^{-3}$ and $\epsilon = 10^{-8}$ [10]. The gradient of the loss function is reported in **Supplementary Methods in S1 File**.

The ACMTF framework of putting the norm of a component into $\Lambda$ and constraining the loading vectors to become norm one, allows the $\Lambda$ matrix to encode the Common, Local, and Distinct (CLD) structure of the data (**Fig 1**). For example, the CLD-structure of a three-block case $\underline{X}^{(1)}$, $\underline{X}^{(2)}$, $\underline{X}^{(3)}$ can be encoded in $\Lambda$ through,

$$\Lambda = \begin{matrix} & \overset{C_{123}}{} & \overset{L_{12}}{} & \overset{L_{13}}{} & \overset{L_{23}}{} & \overset{D_1}{} & \overset{D_2}{} & \overset{D_3}{} \\ \underline{X}^{(1)} \\ \underline{X}^{(2)} \\ \underline{X}^{(3)} \end{matrix} \begin{bmatrix} \lambda_{11} & \lambda_{12} & \lambda_{13} & 0 & \lambda_{15} & 0 & 0 \\ \lambda_{21} & \lambda_{22} & 0 & \lambda_{24} & 0 & \lambda_{26} & 0 \\ \lambda_{31} & 0 & \lambda_{33} & \lambda_{34} & 0 & 0 & \lambda_{37} \end{bmatrix}$$

(5)

where $\lambda_{pf} > 0$ corresponds to data block $p$ contributing to component $f$, 0 corresponds to the data block not contributing to a component, $C_{123}$ indicates a common component shared between all blocks, $L_{12}$, $L_{13}$ and $L_{23}$ indicate local components shared between some blocks, and $D_1$, $D_2$ and $D_3$ indicate distinct components unique to one block. Sparsity is then achieved by applying an $L_1$ penalty on the $\Lambda$-matrix. While this example uses 7 components to showcase the entire CLD-structure, multiple components may be needed to fully describe a specific type of variation in real data (e.g., two components for $C_{123}$) and some types of variation may be absent altogether.

### Advanced Coupled Matrix and Tensor Factorization Regression (ACMTF-R)

We present Advanced Coupled Matrix and Tensor Factorization Regression (ACMTF-R) to describe common, local, and distinct variation of interest in the data blocks in the context of a dependent variable $y$ by adding a regression term and a tuning parameter $\pi$ to the loss function. The introduction of a tuning parameter has been used successfully in other approaches such as Principal Covariates Regression (PCovR) [24,26] and Multiway Covariates Regression (MCovR) [25] to steer the solution towards explaining the data blocks or towards predicting $y$. This gives the user more information about how the addition of $y$ affects the model of the data. We only define ACMTF-R in the case where the first mode is shared across all data blocks and $y$. Hence $y$ is a column vector of size $I \times 1$. The loss function of ACMTF-R is then defined as

$$f\left(\alpha,\beta,\varepsilon,\pi,\Lambda,\boldsymbol{A},\boldsymbol{B}^{(1)},\boldsymbol{C}^{(1)},\ldots,\boldsymbol{B}^{(P)},\boldsymbol{C}^{(P)}\right) = \pi \sum_{p=1}^{P} \left\| \underline{\boldsymbol{X}}^{(p)} - \underline{\hat{\boldsymbol{X}}}^{(p)} \right\|^2$$
$$+ (1-\pi)\left\| \boldsymbol{y} - \boldsymbol{A}\rho \right\|^2$$
$$+ \beta \sum_{p=1}^{P}\sum_{f=1}^{F} \sqrt{\lambda_{pf}^2 + \epsilon}$$
$$+ \alpha \sum_{p=1}^{P}\sum_{f=1}^{F} \left( \|\boldsymbol{a}_f\| - 1 \right)^2$$
$$+ \alpha \sum_{p=1}^{P}\sum_{f=1}^{F} \left( \left\| \boldsymbol{b}_f^{(p)} \right\| - 1 \right)^2$$
$$+ \alpha \sum_{p=1}^{P}\sum_{f=1}^{F} \left( \left\| \boldsymbol{c}_f^{(p)} \right\| - 1 \right)^2$$

$$(6)$$

where $\pi$ is a tuning parameter used to focus the model on explaining the data blocks $\underline{\boldsymbol{X}}^{(1)}$, $\underline{\boldsymbol{X}}^{(2)}$, ..., $\underline{\boldsymbol{X}}^{(P)}$ versus predicting $\boldsymbol{y}$, and $\rho$ is the $F \times 1$ vector of regression coefficients of the factor matrix of the first mode $\boldsymbol{A}$ onto $\boldsymbol{y}$. Here $\rho$ is given when $\boldsymbol{A}$ is defined, through

$$\rho = \boldsymbol{A}^+ \boldsymbol{y}. \tag{7}$$

The gradient of the loss function is reported in **Supplementary Methods in S1 File**.

### Prediction of $y$ using a new sample

A fitted ACMTF-R model can be used to predict $y_{new}$ for a new sample using a similar procedure as Latent Root Regression [39,40]. First, this requires a brief description of obtaining the subject mode loadings for a new sample in CANDECOMP/PARAFAC (CP) [34,36,41,42]. Subsequently, this result is extended to ACMTF-R, after which the prediction step is described.

In the CP case, finding the subject mode loadings $\boldsymbol{a}_{new}$ with size $F \times 1$ for a sample $\boldsymbol{X}_{new}$ with size $J \times K$ requires solving the problem:

$$\underset{\boldsymbol{a}_{new}}{\text{argmin}} \left\| \boldsymbol{X}_{new} - \boldsymbol{B}\text{diag}(\boldsymbol{a}_{new})\boldsymbol{C}^T \right\|_F^2 = \underset{\boldsymbol{a}}{\text{argmin}} \left\| \text{vec}(\boldsymbol{X}_{new}) - (\boldsymbol{C} \odot \boldsymbol{B})\boldsymbol{a}_{new} \right\|_F^2 \tag{8}$$

where $\boldsymbol{B}$ and $\boldsymbol{C}$ are the feature and time mode loadings of a previously fitted model [36]. This problem statement comes down to the least squares solution:

$$\boldsymbol{a}_{new} = (\boldsymbol{C} \odot \boldsymbol{B})^+ \text{vec}(\boldsymbol{X}_{new}) = \boldsymbol{Z}^+ \text{vec}(\boldsymbol{X}_{new}) \tag{9}$$

where the block matrix $\boldsymbol{Z} = \boldsymbol{C} \odot \boldsymbol{B}$ with size $JK \times F$ is used for convenience and where $\boldsymbol{Z}^+$ is the Moore-Penrose inverse of $\boldsymbol{Z}$.

In the case of ACMTF-R, the shared subject mode loadings across the blocks need to be identified. This is done by finding the block matrix $\boldsymbol{Z}^{(p)}$ with size $K^{(p)}J^{(p)} \times F$ through

$$\boldsymbol{Z}^{(p)} = \lambda_p^T \odot \boldsymbol{C}^{(p)} \odot \boldsymbol{B}^{(p)} \tag{10}$$

which can be concatenated across all blocks into one matrix for all data blocks $1, \ldots, P$ simultaneously as

$$\boldsymbol{Z} = \begin{pmatrix} \boldsymbol{Z}^{(1)} \\ \boldsymbol{Z}^{(2)} \\ \vdots \\ \boldsymbol{Z}^{(P)} \end{pmatrix} \tag{11}$$

where $\boldsymbol{Z}$ is of size $\left( K^{(1)} J^{(1)} + K^{(2)} J^{(2)} + \ldots + K^{(P)} J^{(P)} \right) \times F$. The shared subject mode loadings $\boldsymbol{a}_{new}$ with size $F \times 1$ for a new sample $\boldsymbol{X}_{new}$ can then be found by reusing Equation 10.

Finally, the prediction of $y_{new}$ is done using the ACMTF-R regression coefficients $\rho$ through

$$y_{new} = \rho^T \boldsymbol{a}_{new}. \tag{12}$$

When multiple new samples are obtained, this procedure is performed per sample. The procedure is performed by the npred() function in the supplied R package.

## Implementation and stopping criteria

The decomposition of ACMTF is achieved through an all-at-once optimization algorithm, originally developed in [15] for MATLAB, and now available in the CMTFtoolbox package for R on CRAN (development version on GitHub and https://doi.org/10.5281/zenodo.16633119). In ACMTF-R, this algorithm is modified in two ways: (1) the loss and gradient of the all-at-once optimization step is modified to include the tuning between data reconstruction and prediction of $\boldsymbol{y}$ (**Supplementary Methods in S1 File**), and (2) the regression coefficients ($\rho$) are found using a closed-form solution in a separate step. The all-at-once optimisation is achieved by defining the loss and gradient function and subsequently using the nonlinear conjugate gradient (NCG) method with Hestenes-Stiefel updates and the Moré-Thuente line search as originally suggested [15,18]. For the provided R package, this was implemented using the mize package (v0.2.4, [43]). The provided package also supports the Limited-memory Broyden-Fletcher-Goldfarb-Shanno algorithm (L-BFGS) to speed up the line search as an experimental feature, but this setting is not used for any results in this paper [44–46].

ACMTF-R uses the same default values as ACMTF for the minimum value of the loss function (default $10^{-10}$) and the minimum change in loss function between two function evaluations (default $10^{-10}$) as stopping criteria [37]. The all-at-once optimization implementation through the mize package also defines three other stopping criteria: the maximum number of iterations to calculate (default 10 000), the maximum number of function evaluations that are allowed (default 10 000), the minimum l2-norm of the gradient vector (default $10^{-10}$), and the absolute value of the size of the parameter update (default $10^{-10}$). Fitting an ACMTF-R model for any of the datasets in this paper takes less than a minute of computational time on an average computer (Microsoft Windows 11 Home v10.0.22631 with an 12<sup>th</sup> Gen Intel® Core™ i5-12400F running 2500 MHz with 6 cores or 12 logical processors and 16 Gb RAM) and is comparable to the computational cost of ACMTF due to the closed-form regression step. Multi-core parallelisation has been implemented as part of the provided R package to allow many randomly initialised models to be fitted simultaneously. This is needed to efficiently find the appropriate number of components through cross-validation, as well as to find the global minimum for a given number of components.

## Simulation approach

We demonstrate the effect of the ACMTF-R model parameters using various simulations (**Fig 2**; code supplied at https://doi.org/10.5281/zenodo.16633167). We follow the guidelines provided in previous work [10]. We generate factor matrices $\boldsymbol{A} \in \mathbb{R}^{I \times F}$, $\boldsymbol{B}^{(1)} \in \mathbb{R}^{J^{(1)} \times F}$, $\boldsymbol{C}^{(1)} \in \mathbb{R}^{K^{(1)} \times F}$, $\boldsymbol{B}^{(2)} \in \mathbb{R}^{J^{(2)} \times F}$, $\boldsymbol{C}^{(2)} \in \mathbb{R}^{K^{(2)} \times F}$, $\boldsymbol{B}^{(3)} \in \mathbb{R}^{J^{(3)} \times F}$ and $\boldsymbol{C}^{(3)} \in \mathbb{R}^{K^{(3)} \times F}$ with random entries drawn from the standard normal distribution $N \sim (0, 1)$. The columns of the factor matrices are normalized to unit norm. We set $I = 100$, $J^{(1)} = J^{(2)} = J^{(3)} = 200$, and $K^{(1)} = K^{(2)} = K^{(3)} = 5$ to resemble a typical pre-processed dataset (for example, a mixture of microbiome and metabolomics data). The factor matrices are used to create three third-order tensors $\underline{\boldsymbol{X}}^{(1)} \in \mathbb{R}^{100 \times 200 \times 5}$, $\underline{\boldsymbol{X}}^{(2)} \in \mathbb{R}^{100 \times 200 \times 5}$, $\underline{\boldsymbol{X}}^{(3)} \in \mathbb{R}^{100 \times 200 \times 5}$ through

$$\underline{\boldsymbol{X}}^{(1)} = \left[\!\left[ \lambda_1; \boldsymbol{A}, \boldsymbol{B}^{(1)}, \boldsymbol{C}^{(1)} \right]\!\right],$$

$$\underline{\boldsymbol{X}}^{(2)} = \left[\!\left[ \lambda_2; \boldsymbol{A}, \boldsymbol{B}^{(2)}, \boldsymbol{C}^{(2)} \right]\!\right],$$

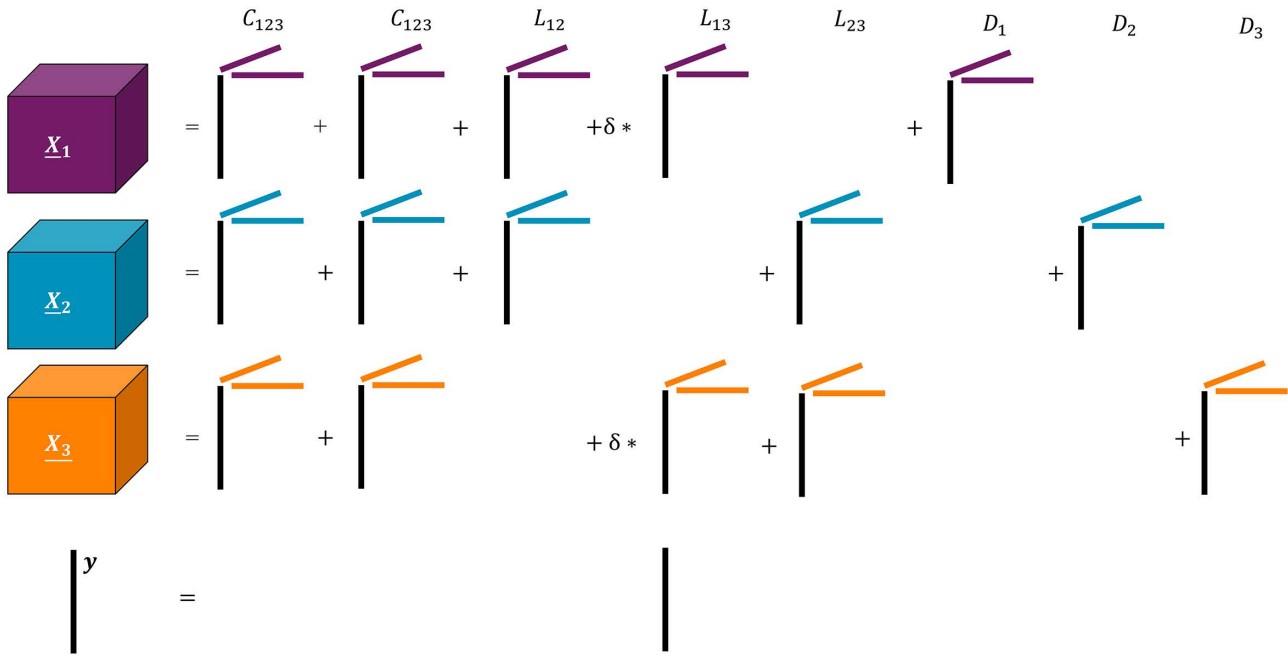

**Fig 2. Overview of the simulation setup of the study.** All simulations contain a CLD-structure of 2 global components shared between blocks $\underline{\mathbf{X}}^{(1)}, \underline{\mathbf{X}}^{(2)}, \underline{\mathbf{X}}^{(3)}$, 3 local components, and a distinct component unique to each block. This yields 8 components total for ACMTF-R to identify. All components are norm 1, except for the second local component whose norm $\delta$ varies per simulation. **y** is equal to the subject mode loadings of the second local component. Noise and terms corresponding to components that have no contribution to a block have been removed for visual clarity. Subject mode loadings (black lines) are equal between the data blocks per component but are different between components.

and

$$\underline{\mathbf{X}}^{(3)} = \left[\left[\lambda_3; \mathbf{A}, \mathbf{B}^{(3)}, \mathbf{C}^{(3)}\right]\right].$$

where we encode the required CLD-structure into $\mathbf{\Lambda}$ consisting of eight components total: two common components shared across all blocks, three local components, and a distinct component for each block. This corresponds to the $\mathbf{\Lambda}$-matrix:

$$\mathbf{\Lambda} = \begin{bmatrix} 1 & 1 & 1 & \delta & 0 & 1 & 0 & 0 \\ 1 & 1 & 1 & 0 & 1 & 0 & 1 & 0 \\ 1 & 1 & 0 & \delta & 1 & 0 & 0 & 1 \end{bmatrix}$$

where all components are norm 1, except for the second local component whose size $\delta$ varies per simulation.

Next, randomly distributed noise is added to the third-order tensors through

$$\underline{\mathbf{X}}^{(p)}_{noisy} = \underline{\mathbf{X}}^{(p)} + \eta_X \frac{\left\|\underline{\mathbf{X}}^{(p)}\right\|}{\left\|\underline{\mathbf{N}}^{(p)}\right\|} \underline{\mathbf{N}}^{(p)} \tag{13}$$

where $\eta_X$ indicates the noise level, which is the same across all blocks.

We generate **y** equal to the subject mode loadings of the second local component:

$$\mathbf{y} = \mathbf{a}_{L12}$$

and some noise is added through

$$\boldsymbol{y}_{noisy} = \boldsymbol{y} + \eta_y \frac{\|\boldsymbol{y}\|}{\|\boldsymbol{n}_y\|} \boldsymbol{n_y} \tag{14}$$

where $\eta_y$ indicates the noise level on $\boldsymbol{y}$. The outcome $\boldsymbol{y}_{noisy}$ is then normalized to norm 1.

## Simulation 1: ACMTF-R can detect a small, hidden component by leveraging $\boldsymbol{y}$

In the first simulation, the ability of ACMTF-R to detect a small, hidden component by leveraging $\boldsymbol{y}$ was examined. This was done using a three-block simulation with $\delta = 0.03$, 40% noise on $\underline{\boldsymbol{X}}^{(1)}, \underline{\boldsymbol{X}}^{(2)}, \underline{\boldsymbol{X}}^{(3)}$ ($\eta_X = 0.40$), and 5% noise on $\boldsymbol{y}$ ($\eta_y = 0.05$). The data were then block scaled to Frobenius norm 1. One hundred randomly initialised eight-component ACMTF models were compared with one hundred randomly initialised eight-component ACMTF-R models using different values of $\pi \in \{0.1, 0.2, \ldots, 0.9\}$, keeping all other parameter and convergence settings at their default values. Subsequently, factor recovery was assessed using Factor Match Score and Tucker Congruence Coefficient. The line search algorithm in most ACMTF-R models was terminated due to the absolute change for the size of the parameter update stopping criterion, while ACMTF was terminated only due to the minimum value of the loss function being reached (**Supplementary Figure 1 in S1 File**).

## Simulation 2: The benefit of supervision using $\boldsymbol{y}$ versus the size of the hidden component

In the second simulation, the improvement of ACMTF-R compared to ACMTF was assessed by changing the size of the small, hidden component. This was done using:

$$\delta \in \{0.01, 0.02, 0.03, 0.04, 0.05, 0.1, 0.2, 0.3, 0.4, 0.5\}.$$

The noise on $\underline{\boldsymbol{X}}^{(1)}, \underline{\boldsymbol{X}}^{(2)}, \underline{\boldsymbol{X}}^{(3)}$ was set at 40% ($\eta_X = 0.40$), and the noise on $\boldsymbol{y}$ was 5% ($\eta_y = 0.05$). The data were then block scaled to Frobenius norm 1. One hundred randomly initialised eight-component ACMTF models were compared with one hundred randomly initialised eight-component ACMTF-R models using different values of $\pi \in \{0.1, 0.2, \ldots, 0.9\}$ for each case, keeping all other parameter and convergence settings at their default values. Subsequently, factor recovery was assessed using Factor Match Score and Tucker Congruence Coefficient. The line search algorithm in most ACMTF-R models was terminated due to the absolute change for the size of the parameter update stopping criterion, while most ACMTF models were terminated only due to the minimum value of the loss function being reached (**Supplementary Figure 2 in S1 File**).

## Simulation 3: The benefit of supervision using $\boldsymbol{y}$ versus noise on X

In the third simulation, the improvement of ACMTF-R compared to ACMTF was assessed by changing the amount of noise on $\underline{\boldsymbol{X}}^{(1)}, \underline{\boldsymbol{X}}^{(2)}, \underline{\boldsymbol{X}}^{(3)}$. This was done using $\delta = 0.03$ and a variable amount of noise on $\underline{\boldsymbol{X}}^{(1)}, \underline{\boldsymbol{X}}^{(2)}, \underline{\boldsymbol{X}}^{(3)}$,

$$\eta_X \in \{0.0, 0.1, 0.2, 0.3, 0.4, 0.5, 0.6, 0.7, 0.8, 0.9, 1, 2, 3, 4, 5, 10, 50, 100\},$$

and the noise on $\boldsymbol{y}$ was 5% ($\eta_y = 0.05$). The data were then block scaled to Frobenius norm 1. One hundred randomly initialised eight-component ACMTF models were compared with one hundred randomly initialised eight-component ACMTF-R models using different values of $\pi \in \{0.1, 0.2, \ldots, 0.9\}$ for each case, keeping all other parameter and convergence settings at their default values. Subsequently, factor recovery was assessed using Factor Match Score and Tucker Congruence Coefficient. For most noise levels ($\eta_X < 2$), the line search algorithm in most ACMTF-R models was terminated due to the absolute change for the size of the parameter update stopping criterion, while most ACMTF models

were terminated only due to the minimum value of the loss function being reached (**Supplementary Figure 3 in S1 File**). At higher noise levels ($\eta_X > 2$), most ACMTF models were terminated due to the relative change in the loss function being reached.

### Simulation 4: The benefit of supervision using *y* versus noise on *y*

In the fourth and final simulation, the improvement of ACMTF-R compared to ACMTF was assessed by changing the amount of noise on *y*. This is done using the three-block simulation with $\delta = 0.03$, with 40% noise on $\underline{X}^{(1)}, \underline{X}^{(2)}, \underline{X}^{(3)}$ ($\eta_X = 0.40$), and a variable amount of noise on *y*:

$$\eta_y \in \{0.0, \ 0.05, \ 0.1, \ 0.2, \ 0.3, \ 0.4, \ 0.5, \ 0.6, \ 0.7, \ 0.8, \ 0.9, \ 1, \ 2, \ 3, \ 4, \ 5, \ 10, \ 50, \ 100\}.$$

The data were then block scaled to Frobenius norm 1. One hundred randomly initialised eight-component ACMTF models were compared with one hundred randomly initialised eight-component ACMTF-R models using different values of $\pi \in \{0.1, \ 0.2, \ \ldots, 0.9\}$ for each case, keeping all other parameter and convergence settings at their default values. Subsequently, factor recovery was assessed using Factor Match Score and Tucker Congruence Coefficient. The line search algorithm in most ACMTF-R models was terminated due to the absolute change for the size of the parameter update stopping criterion, while ACMTF was terminated only due to the minimum value of the loss function being reached (**Supplementary Figure 4 in S1 File**).

### Simulation evaluation – Tucker Congruence Coefficient

In this paper, the recovered factors are compared against the input factors of the simulation using the absolute Tucker Congruence Coefficient (TCC) [47,48] through

$$\left| \varphi\left(\boldsymbol{x}, \boldsymbol{y}\right) \right| = \left| \frac{\sum_i x_i y_i}{\sqrt{\sum_i x_i^2 \sum_i y_i^2}} \right| \tag{15}$$

where $\boldsymbol{x}$ and $\boldsymbol{y}$ are the recovered and true factors, respectively. The TCC can be interpreted as the cosine of the angle between $\boldsymbol{x}$ and $\boldsymbol{y}$. A TCC value in the range of 0.85–0.94 suggests that the recovered and input factors are fairly similar, whereas a value of 0.95–1 means that they are essentially equal [49]. Although TCC is insensitive to scalar multiplication of the vectors (that is, $\varphi\left(\boldsymbol{x}, \boldsymbol{y}\right) = \varphi(a\boldsymbol{x}, b\boldsymbol{y})$ for any non-zero scalars $a$ and $b$), it is sensitive to sign flips between $\boldsymbol{x}$ and $\boldsymbol{y}$ [49]. For this reason, we use the absolute value instead.

### Simulation evaluation – Factor Match Score

The Factor Match Score (FMS) is an adaptation of the Tucker Congruence Coefficient and indicates the similarity of two multi-way models across all components. For model evaluation, FMS was used to compare the decomposition against the input factors. This depends on the CLD-type of the relevant factor. For the hidden local component $L_{13}$ in the simulations, $FMS_{L13}$ is defined as

$$FMS_{L13} = \frac{\left| \boldsymbol{a}_{L13}^T \widetilde{\boldsymbol{a}}_{L13} \right|}{\|\boldsymbol{a}_{L13}\| \|\widetilde{\boldsymbol{a}}_{L13}\|} \frac{\left| \left[\boldsymbol{b}_{L13}^{(1)}\right]^T \widetilde{\boldsymbol{b}}_{L13}^{(1)} \right|}{\left\| \boldsymbol{b}_{L13}^{(1)} \right\| \left\| \widetilde{\boldsymbol{b}}_{L13}^{(1)} \right\|} \frac{\left| \left[\boldsymbol{c}_{L13}^{(1)}\right]^T \widetilde{\boldsymbol{c}}_{L13}^{(1)} \right|}{\left\| \boldsymbol{c}_{L13}^{(1)} \right\| \left\| \widetilde{\boldsymbol{c}}_{L13}^{(1)} \right\|} \frac{\left| \left[\boldsymbol{b}_{L13}^{(3)}\right]^T \widetilde{\boldsymbol{b}}_{L13}^{(3)} \right|}{\left\| \boldsymbol{b}_{L13}^{(3)} \right\| \left\| \widetilde{\boldsymbol{b}}_{L13}^{(3)} \right\|} \frac{\left| \left[\boldsymbol{c}_{L13}^{(3)}\right]^T \widetilde{\boldsymbol{c}}_{L13}^{(3)} \right|}{\left\| \boldsymbol{c}_{L13}^{(3)} \right\| \left\| \widetilde{\boldsymbol{c}}_{L13}^{(3)} \right\|} \tag{16}$$

where $\boldsymbol{a}_{L13}$, $\boldsymbol{b}_{L13}$, and $\boldsymbol{c}_{L13}$ contain the true subject, feature and time loadings of component $L_{13}$, and $\widetilde{\boldsymbol{a}}_{L13}$, $\widetilde{\boldsymbol{b}}_{L13}$, and $\widetilde{\boldsymbol{c}}_{L13}$ are the best matching subject, feature and time loadings in the fitted model. The FMS of the other components are defined equivalently. In the simulations, the best matching component is found using the Hungarian algorithm [50,51] for

all pairwise combinations of components such that the FMS is maximised and the expected CLD structure is recovered (**Supplementary Methods in S1 File**). The FMS is a value in the range [0,1], where $FMS = 1$ corresponds to the model correctly recovering the input loadings.

The recovery of all input loadings can be quantified through $FMS_{overall}$, defined as

$$FMS_{overall} = \frac{FMS_{C123} + FMS_{C123} + FMS_{L12} + FMS_{L13} + FMS_{L23} + FMS_{D1} + FMS_{D2} + FMS_{D3}}{8} \tag{17}$$

where $FMS_{overall} = 1$ corresponds to all input components being recovered correctly. Recovery of the CLD structure is assessed using the Lambda Similarity Index, which is defined and reported in the **Supplementary Materials in S1 File**.

## Example dataset: Jakobsen2025

We include an application study of ACMTF-R on longitudinally measured human milk (HM) microbiome, HM metabolomics and infant faecal microbiome data. Full details on the preprocessing of the data is outlined in the original paper [52] and preprocessing of longitudinal compositional data is detailed in [53].

Briefly, the zOTU data and taxonomic information of the infant faecal microbiome and human milk microbiome were processed in R (v4.4.1, [54]). For the infant faecal microbiome data, features were kept if they had ≤ 75% sparsity across the dataset. This step resulted in 93 out of 565 features being selected. For the human milk microbiome data, features were kept if they had ≤ 85% sparsity across the dataset. This step resulted in 115 out of 707 features being selected. We performed a centred log-ratio transformation with a pseudo-count of 1 to correct for compositionality [55,56]. Subsequently the data was converted to a three-way tensor, keeping missing samples as a row of NAs. This resulted in three-way arrays of size 160 subjects × 93 microbial taxa × 3 time points and 169 subjects × 115 microbial taxa × 4 time points containing microbial abundances for the infant faecal and human milk microbiomes, respectively.

The human milk (HM) metabolomics data were processed using R (v4.4.1, [54]). Values below the detection limit were imputed with a random value between 0 and the detection limit per metabolite to preserve their distribution. Next, the dataset was (natural) log transformed to stabilise the variance. The dataset was then converted to a three-way tensor of size 165 subjects × 70 metabolites × 4 time points.

Since ACMTF-R assumes that the subject mode is shared between all data blocks, we took the intersection of the subject identifiers across all three datasets and removed the other subjects. This homogenised the subject mode size $I$ across all data blocks to 158 subjects. The data were then centred across the subject mode such that variation per time point is only due to between-subject variation, and scaled within the feature mode to make all features equally important for the modelling procedure [57,58]. As a result, the models will focus on inter-subject differences and the features involved, while the time mode loadings indicate when these differences are largest.

After centring and scaling, the datasets were block scaled to Frobenius norm 1 to equalise the variance across the blocks. The response variable $\boldsymbol{y}$ of size $158 \times 1$ contained the maternal pre-pregnancy BMI for all mother-infant dyads, which was centred and subsequently scaled to norm 1 prior to modelling. ACMTF-R models were then created using the appropriate number of components and regressed on maternal pre-pregnancy BMI.

## Selecting the appropriate number of ACMTF components

The optimal number of components does not exist in real multi-omics applications, since this depends on data idiosyncrasies as well as biological interpretability of the model. Therefore, this paper presents a data analyst guided triangulation approach based on model stability, reproducibility, fit, degeneracy, and biological interpretability to identify the appropriate number of components for ACMTF. This is done through a K-fold cross-validation scheme, as well as a random initialization scheme. The full procedure is available as the ACMTF_modelSelection() function in the supplied R package.

In the K-fold cross-validation scheme, one fold of the subjects is withheld for all blocks as test set data, while randomly initialised ACMTF models are fitted on the training data using $f$ components, keeping all other model parameters ($\alpha$, $\beta$, $\epsilon$) fixed. To avoid data leakage, the means, standard deviations, and norms that are removed from the training data are also removed from the test set data. Model stability is then reported by pairwise comparing the models with the lowest loss per fold and calculating the $FMS_{CV}$, defined as

$$FMS_{CV} = \frac{1}{F} \sum_{f=1}^{F} \frac{\left|\boldsymbol{b}_f^T \widetilde{\boldsymbol{b}}_f\right|}{\|\boldsymbol{b}_f\|\|\widetilde{\boldsymbol{b}}_f\|} \frac{\left|\boldsymbol{c}_f^T \widetilde{\boldsymbol{c}}_f\right|}{\|\boldsymbol{c}_f\|\|\widetilde{\boldsymbol{c}}_f\|}$$

(18)

where $\boldsymbol{b}_f$, and $\boldsymbol{c}_f$ contain the feature and time loadings for component $f$ in one model, $\widetilde{\boldsymbol{b}}_f$, and $\widetilde{\boldsymbol{c}}_f$ are the best matching subject, feature and time loadings for component $f$ in a second model, and the subject mode term is eliminated from the calculated due to not being comparable between CV folds [15,59]. In the application, the matching of components is achieved using the Hungarian algorithm [50,51] for all pairwise combinations of components such that the FMS is maximised (**Supplementary Methods in S1 File**). Hence, the distribution of FMS values across all pairwise comparisons of CV (or jack-knifed) folds gives an indication of the stability of the decomposition for a given number of components $F$. Previous work has established that 95% of folds should have $FMS_{CV} \geq 0.95$ for three-way data blocks and $FMS_{CV} \geq 0.9$ for two-way data blocks in the ACMTF case for the decomposition to be considered replicable [59]. The appropriate number of components is expected to have a stable $FMS_{CV}$ near one across all comparisons.

In the second arm of the approach, randomly initialised ACMTF models are fitted to the full data using $F$ components. Model sensitivity to the starting values and to local minima is then assessed by pairwise comparing the models and calculating the $FMS_{random}$ through

$$FMS_{random} = \frac{1}{F} \sum_{f=1}^{F} \frac{\left|\boldsymbol{a}_f^T \widetilde{\boldsymbol{a}}_f\right|}{\|\boldsymbol{a}_f\|\|\widetilde{\boldsymbol{a}}_f\|} \frac{\left|\boldsymbol{b}_f^T \widetilde{\boldsymbol{b}}_f\right|}{\|\boldsymbol{b}_f\|\|\widetilde{\boldsymbol{b}}_f\|} \frac{\left|\boldsymbol{c}_f^T \widetilde{\boldsymbol{c}}_f\right|}{\|\boldsymbol{c}_f\|\|\widetilde{\boldsymbol{c}}_f\|}$$

(19)

where $\boldsymbol{a}_f$, $\boldsymbol{b}_f$, and $\boldsymbol{c}_f$ contain the subject, feature and time loadings for component $f$ in one model, and $\widetilde{\boldsymbol{a}}_f$, $\widetilde{\boldsymbol{b}}_f$, and $\widetilde{\boldsymbol{c}}_f$ are the best matching subject, feature and time loadings for component $f$ in a second model [15,59]. For the application, the matching of components is again achieved using the Hungarian algorithm [50,51] for all pairwise combinations of components such that the FMS is maximised (**Supplementary Methods in S1 File**). The distribution of FMS values across all pairwise comparisons of randomly initialised models for a given number of components $F$ gives an indication of the stability of the decomposition. The appropriate number of components is expected to have a stable $FMS_{random}$ near one across all randomly initialised models.

Additionally, it may be possible for ACMTF and ACMTF-R to split up common or local components into distinct components with highly similar subject mode loadings when too many components are chosen. This phenomenon can be detected by pairwise comparing all columns in the subject mode per model and calculating the maximum absolute Tucker Congruence Coefficient $\phi$ [47–49]. We define this as the Degeneracy Score, which is computed through

$$Degeneracy\ Score = \max_{p \neq q} \left|\phi\left(\boldsymbol{a}_p, \boldsymbol{a}_q\right)\right|$$

(20)

where $p$ and $q$ indicate component numbers of the subject mode within the same model, and $p \neq q$. The Degeneracy Score is in [0, 1], will be close to one when a factor is split into two highly collinear factors, and will have intermediate values otherwise. This procedure is only performed for the randomly initialised ACMTF models since the complete subject mode is needed. The appropriate number of components is expected to have a low Degeneracy Score across all randomly initialised models.

### Selecting the appropriate number of ACMTF-R components

For ACMTF-R, the suggested triangulation approach incorporates a few additional metrics related to the prediction and fit of $y$ for selecting the correct number of components. The full procedure is available as the ACMTFR_modelSelection() function in the supplied R package.

In the cross-validation procedure, the root mean squared error of cross-validation (RMSECV) is calculated by using the models fitted on the training set to predict $y$ in the test set. The appropriate number of components is expected to minimise the RMSECV while having a stable $FMS_{CV}$ near one across all comparisons.

In the random initialisation procedure, the root mean squared error (RMSE) of $y$ and the variance explained in $y$ are reported using the ACMTF-R model with the lowest loss. The appropriate number of components is expected to minimise the RMSE while having a low Degeneracy Score and a stable $FMS_{random}$ near one across all comparisons.

## Results

### Simulation 1: ACMTF-R can detect a small, hidden component by leveraging $y$

We evaluated how the tuning parameter $\pi$ influences the recovery of a local component that is weakly present in some data blocks and strongly reflected in $y$. This was done by applying ACMTF-R to a simulated three-tensor dataset with a known structure containing 2 common, 3 local, and 3 distinct components and 40% noise. All components were norm 1, except for the second local component ($L_{13}$), which was present across $\underline{X}^{(1)}$ and $\underline{X}^{(3)}$ with norm $\delta = 0.03$. Since this component was equal to $y$ with 5% noise, we expected that lowering $\pi$ would enhance its detection. By systematically varying the tuning parameter $\pi$ from 1 (fully focused on explaining the data blocks) to 0 (fully focused on predicting $y$), we examined whether outcome information could reveal a latent structure that would otherwise remain hidden (**Fig 3**). One hundred randomly initialised eight-component ACMTF-R and ACMTF models were created for each setting of $\pi$.

As assessed by the Factor Match Score, the lowest-loss ACMTF-R models using $0.6 \leq \pi \leq 0.9$ showed an improvement in factor recovery for all components compared to the lowest-loss ACMTF model (**Fig 3A**, **Supplementary Figure 5 in S1 File**). This was also the case for factor recovery of the hidden component $L_{13}$ (**Fig 3B**, **Supplementary Figure 6 in S1 File**). Meanwhile, the fraction of models that correctly identified the input CLD structure was comparable to ACMTF (**Supplementary Figure 7-8 in S1 File**).

Next, the lowest-loss ACMTF-R models for each $\pi$ setting were selected for comparison with the lowest-loss ACMTF model (**Fig 3C-3H**). The absolute Tucker Congruence Coefficient ($\varphi$) was used to compare the best matching recovered factor loadings with the input loadings of $L_{13}$ per mode (**Supplementary Methods in S1 File**). This analysis revealed that the recovered shared subject mode (**Fig 3C**) and time mode loadings of $\underline{X}^{(1)}$ (**Fig 3E**) and $\underline{X}^{(3)}$ (**Fig 3G**) were almost identical to the input loadings in the ACMTF-R models for $0.6 \leq \pi \leq 0.9$, while ACMTF failed to identify them. While the recovered feature mode loadings in $\underline{X}^{(1)}$ (**Fig 3D**) and $\underline{X}^{(3)}$ (**Fig 3F**) were not exactly equal to the input factor of the hidden component, their order largely matched the input (Kendall's $\tau = 0.35 - 0.45$ for $0.4 \leq \pi \leq 0.9$; **Supplementary Table 1 in S1 File**).

Taken together, ACMTF-R recovered the hidden component $L_{13}$ across intermediate values of $\pi$ by leveraging $y$, while ACMTF did not.

### Simulation 2: The benefit of supervision using $y$ depends on the size of the hidden component

We next evaluated how the size of the hidden component $L_{13}$ affected the improvement of ACMTF-R compared to ACMTF. This was done using the same three-tensor setup as in the first simulation, except that the size of the second local component ($L_{13}$) was varied between $\delta = 0.01$ to $\delta = 0.5$. We expected the benefit of supervision to decrease as the size of the hidden component $\delta$ increased. As before, the tuning parameter $\pi$ was varied from 0.1 to 0.9 and one hundred randomly initialised eight-component ACMTF-R and ACMTF models were created for each setting (**Fig 4**).

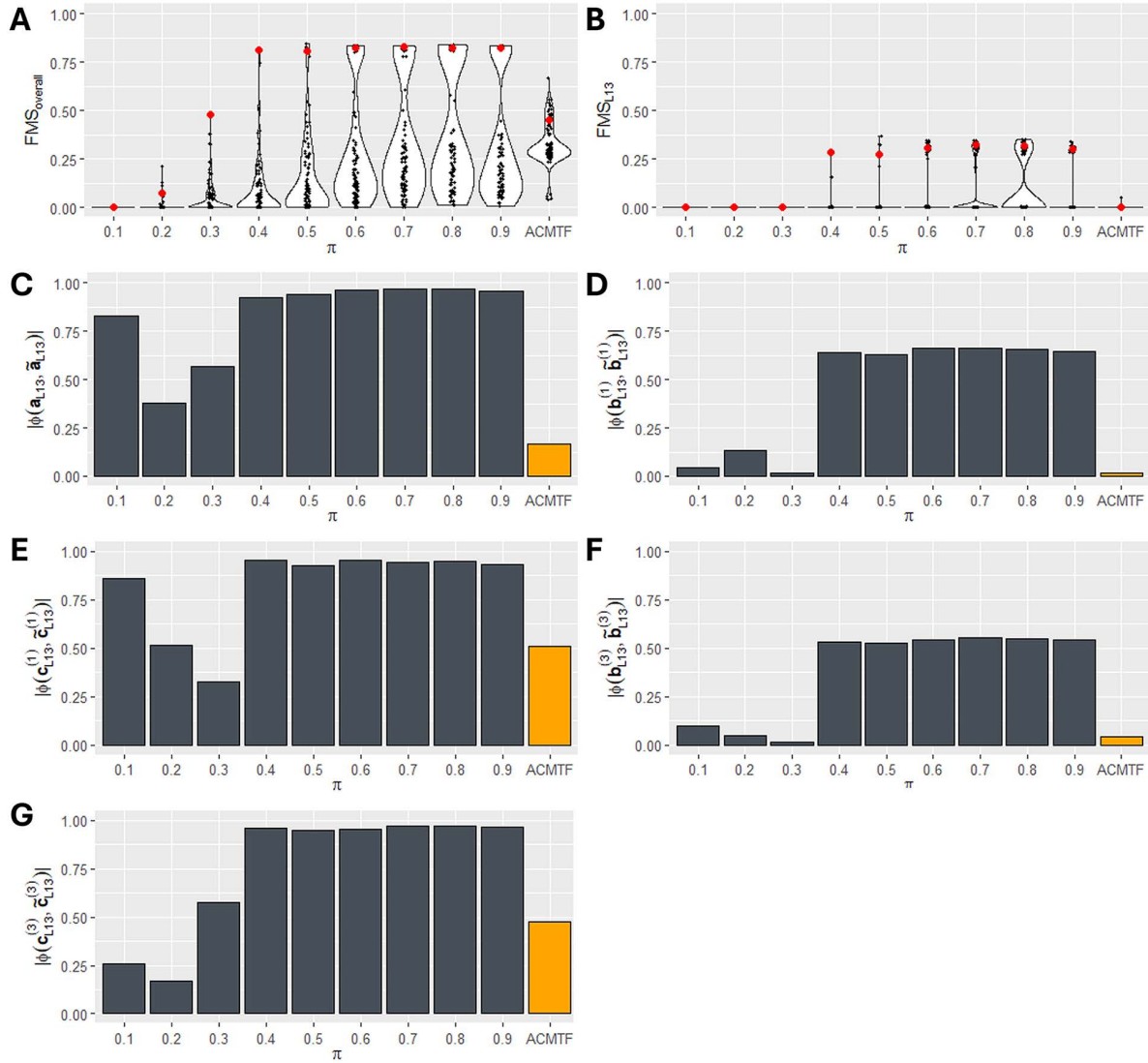

**Fig 3. Recovery of a small, hidden component L$_{13}$ using ACMTF and ACMTF-R.** One hundred randomly initialised ACMTF-R models were fitted per setting of tuning parameter $\pi$. These were compared with one hundred randomly initialised ACMTF models. **(A)** Overall factor recovery, measured using **FMS$_{overall}$**, was higher in ACMTF-R using $0.6 \leq \pi \leq 0.9$ compared to ACMTF. **(B)** Likewise, the FMS$_{L13}$ maximum was higher for ACMTF-R than ACMTF. Red data points in **(A,B)** indicate the model with the lowest loss, which were compared in **(C-H)**. The absolute Tucker Congruence Coefficient ($\varphi$) between the recovered loadings and the input **(C)** subject mode loadings, **(D)** feature mode loadings of $\mathbf{X}^{(1)}$, **(E)** time mode loadings of $\mathbf{X}^{(1)}$, **(F)** feature mode loadings of $\mathbf{X}^{(3)}$, and **(G)** time mode loadings of $\mathbf{X}^{(3)}$ for hidden component L$_{13}$. For all modes, factor recovery using ACMTF-R with intermediate values of $\pi$ was higher compared to ACMTF.

Input factor recovery across all components improved in ACMTF-R compared to ACMTF at $0.4 \leq \pi \leq 0.9$ when the hidden component was small ($0.01 \leq \delta \leq 0.05$), while it deteriorated at lower values of $\pi$ regardless of hidden component size due to overfitting (**Fig 4A**). Meanwhile, factor recovery of the hidden component $L_{13}$ improved in ACMTF-R at $0.4 \leq \pi \leq 0.9$ only at intermediate component sizes ($0.03 \leq \delta \leq 0.05$) (**Fig 4B**). At $\delta > 0.05$, ACMTF and ACMTF-R were both able to identify $L_{13}$, while at $\delta < 0.03$ neither method could recover it (**Supplementary Figure 9 in S1 File**).

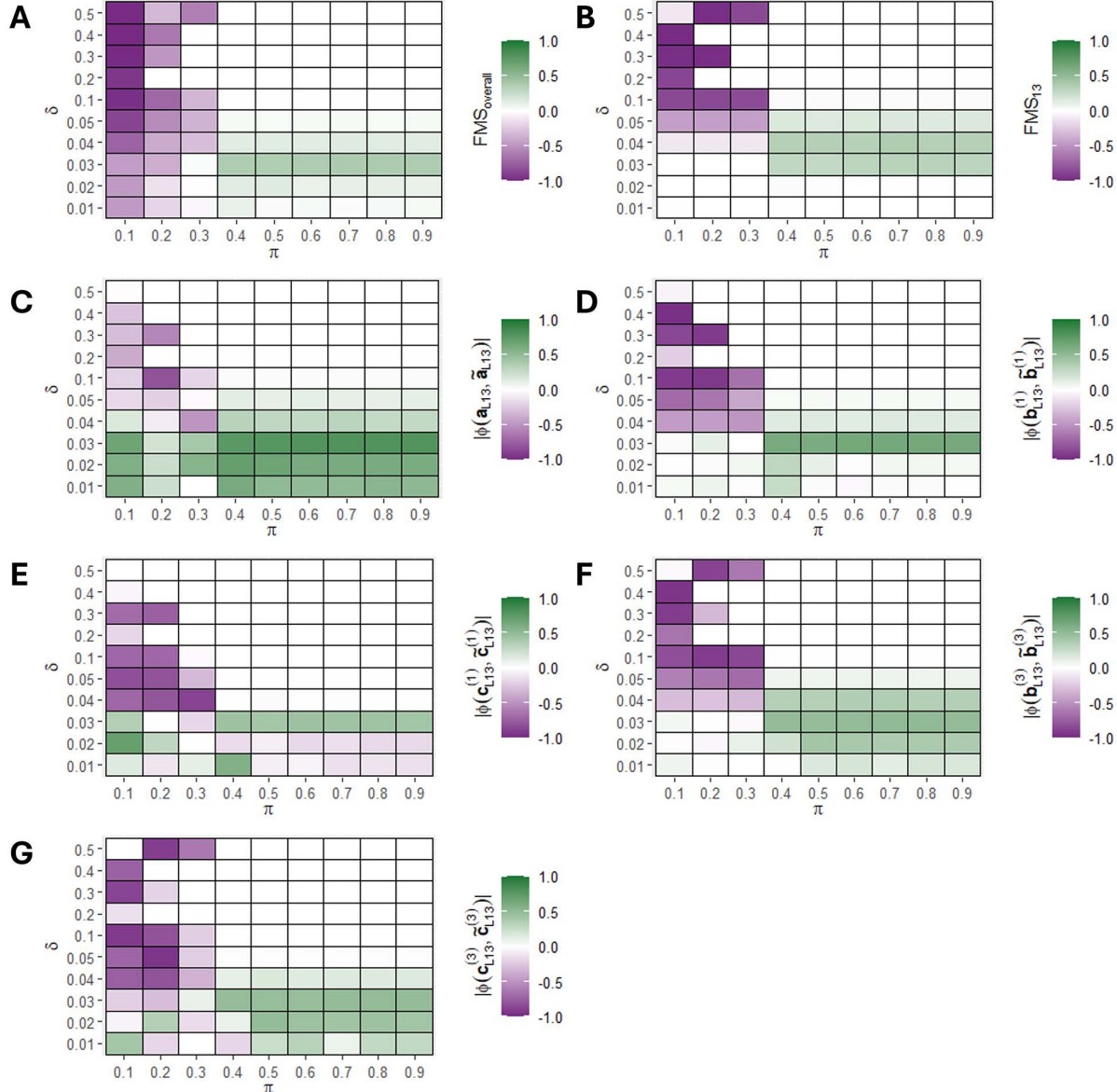

**Fig 4. Improvement in factor recovery for varying sizes of the hidden component $L_{13}$ using ACMTF-R.** One hundred randomly initialised eight-component ACMTF-R and ACMTF models were fitted for each combination of hidden component size ($\delta$) and tuning ($\pi$), after which the lowest-loss models were compared. The improvement in factor recovery is computed by subtracting ACMTF performance from ACMTF-R performance in all panels. Improvement per setting for **(A)** overall factor recovery, **(B)** factor recovery of the hidden component, **(C)** shared subject mode recovery of the hidden component, **(D)** feature mode recovery of $\mathbf{X}^{(1)}$ for the hidden component, **(E)** time mode recovery of $\mathbf{X}^{(1)}$ for the hidden component, **(F)** feature mode recovery of $\mathbf{X}^{(3)}$ for the hidden component, **(G)** time mode recovery of $\mathbf{X}^{(3)}$ for the hidden component. Improvement compared to ACMTF is shown in green, while deterioration is shown in purple. A version of this figure without subtraction of ACMTF performance is shown in **Supplementary Figure 9 in** S1 File.

The recovered shared subject mode loadings of the hidden component showed a consistent recovery at small component sizes (0.01–0.05), regardless of $\pi$ (**Fig 4C**). Meanwhile, the recovery of the feature and time mode loadings improved at similar component sizes only for intermediate values of $\pi$ (0.4–0.9) (**Fig 4D**-**4G**). Taken together, the

simulation revealed that a hidden component that is roughly 1/20$^{th}$ the size of the other components can be successfully recovered by supervising with ACMTF-R using intermediate values of $\pi$.

### Simulation 3: The benefit of supervision using $y$ is consistent for biologically realistic noise levels

We evaluated how the amount of noise on the independent data blocks $\underline{X}^{(1)}$, $\underline{X}^{(2)}$, and $\underline{X}^{(3)}$ affected the improvement in recovering the hidden component $L_{13}$ using ACMTF-R compared to ACMTF. This was done using the same three-tensor setup as in the first simulation, except that the amount of noise on the independent data blocks was varied from $\eta_X = 0.0$ to $\eta_X = 100$. We expected the benefit of supervision to increase as the amount of noise increased. As before, the tuning parameter $\pi$ was varied from 0.1 to 0.9 and one hundred randomly initialised eight-component ACMTF-R and ACMTF models were created for each setting (**Fig 5**).

Input factor recovery across all components improved in ACMTF-R compared to ACMTF at most $\pi$ values (0.3–0.9) at 0−50% noise (**Fig 5A**). However, at 10−30% noise, the improvement was low due to both ACMTF and ACMTF-R recovering the hidden component (**Supplementary Figure 10 in S1 File**). Likewise, factor recovery of the hidden component $L_{13}$ improved only at intermediate noise levels (30−50%) and at intermediate $\pi$ values (0.6–0.9) (**Fig 5B**). When no noise was present, the improvement in factor recovery of the hidden component was zero because both methods were unable to identify it (**Supplementary Figure 10 in S1 File**).

The recovered shared subject mode loadings of the hidden component showed a consistent recovery at all noise levels and values of $\pi$ (**Fig 5C**). The recovery of the feature mode loadings in block $\underline{X}^{(1)}$ improved only at intermediate noise levels (20−50%) and intermediate $\pi$ values (0.6–0.9), while the recovery of the feature mode loadings in block $\underline{X}^{(3)}$ was largely independent of the noise level (**Fig 5D**, **5F**). Surprisingly, the recovery of the time mode loadings in block $\underline{X}^{(1)}$ deteriorated at high noise levels (50 + %), while the recovery of the time mode loadings in block $\underline{X}^{(3)}$ improved at those levels (**Fig 5E**, **5G**). Taken together, this simulation revealed that factor recovery by ACMTF-R improves at biologically realistic noise levels (>30%).

### Simulation 4: The benefit of supervision using $y$ is largely independent of the amount of noise on $y$

We evaluated how the amount of noise on $y$ affects the improvement in recovering the hidden component $L_{13}$ using ACMTF-R compared to ACMTF. This was done using the same three-tensor setup as in the first simulation, except that the amount of noise on $y$ was varied from $\eta_y = 0.0$ to $\eta_y = 100$. We expected the benefit of supervision to decrease as the amount of noise on $y$ increased. As before, the tuning parameter $\pi$ was varied from 0.1 to 0.9 and one hundred randomly initialised eight-component ACMTF-R and ACMTF models were created for each setting (**Fig 6**).

Input factor recovery across all components improved in ACMTF-R compared to ACMTF at intermediate $\pi$ values (0.4–0.9) for most noise levels (0–100%) (**Fig 6A**). Likewise, factor recovery of the hidden component $L_{13}$ improved at intermediate $\pi$ values (0.5–0.9) for most noise levels (0–100%) (**Fig 6B**). At lower $\pi$ values, factor recovery in ACMTF-R deteriorated due to the strong focus on predicting $y$ (**Supplementary Figure 11 in S1 File**). At very high noise levels (100 + %), neither ACMTF nor ACMTF-R were able to recover the hidden component.

The recovered shared subject mode loadings of the hidden component showed a consistent recovery at most noise levels (0−100%) and intermediate values of $\pi$ (0.4–0.9) (**Fig 6C**). Similarly, recovery of the feature and time mode loadings was consistent at most noise levels (0−100%) and intermediate values of $\pi$ (0.4–0.9) (**Fig 6D**-**6G**). Taken together, the simulation revealed that recovery of the hidden component by ACMTF-R is insensitive to biologically realistic noise levels on $y$.

### Application: ACMTF-R enhances multi-omics integration compared to ACMTF and NPLS

To demonstrate the advantage of ACMTF-R for integrating multi-omics data, we analysed a longitudinal dataset comprising human milk (HM) metabolomics, HM microbiome and infant faecal microbiome data, using pre-pregnancy BMI

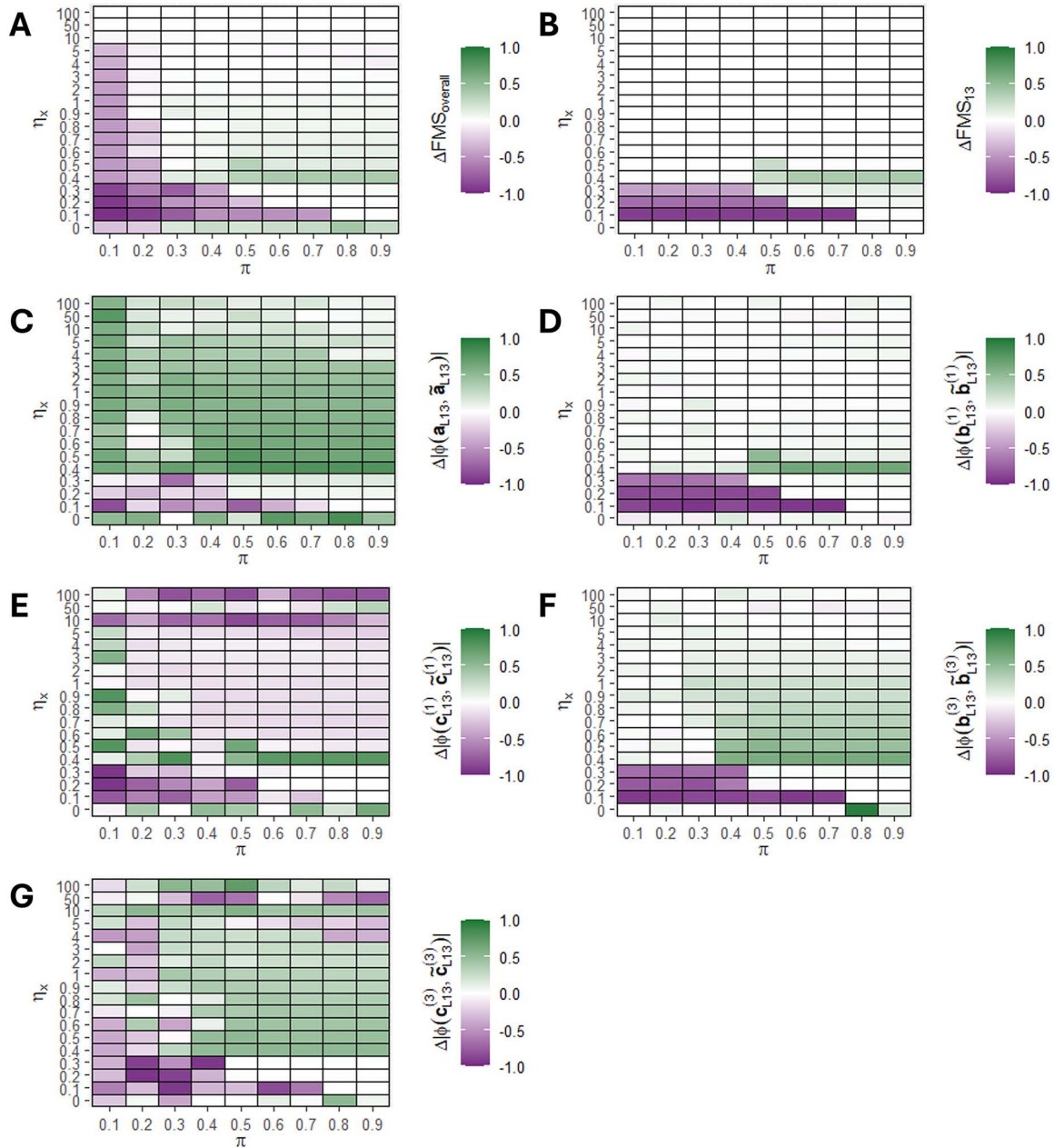

**Fig 5. The improvement in factor recovery by ACMTF-R under different amounts of noise on X.** One hundred randomly initialised eight-component ACMTF-R and ACMTF models were fitted for each combination of noise on X ($\eta_X$) and tuning parameter ($\pi$), after which the lowest-loss models were compared. The improvement in factor recovery is computed by subtracting ACMTF performance from ACMTF-R performance in all panels. Improvement per setting for **(A)** overall factor recovery, **(B)** factor recovery of the hidden component, **(C)** shared subject mode recovery of the hidden component, **(D)** feature mode recovery of $\mathbf{X}^{(1)}$ for the hidden component, **(E)** time mode recovery of $\mathbf{X}^{(1)}$ for the hidden component, **(F)** feature mode recovery of $\mathbf{X}^{(3)}$ for the hidden component, **(G)** time mode recovery of $\mathbf{X}^{(3)}$ for the hidden component. Improvement compared to ACMTF is shown in green, while deterioration is shown in purple. A version of this figure without subtraction of ACMTF performance is shown in **Supplementary Figure 10 in S1 File**.

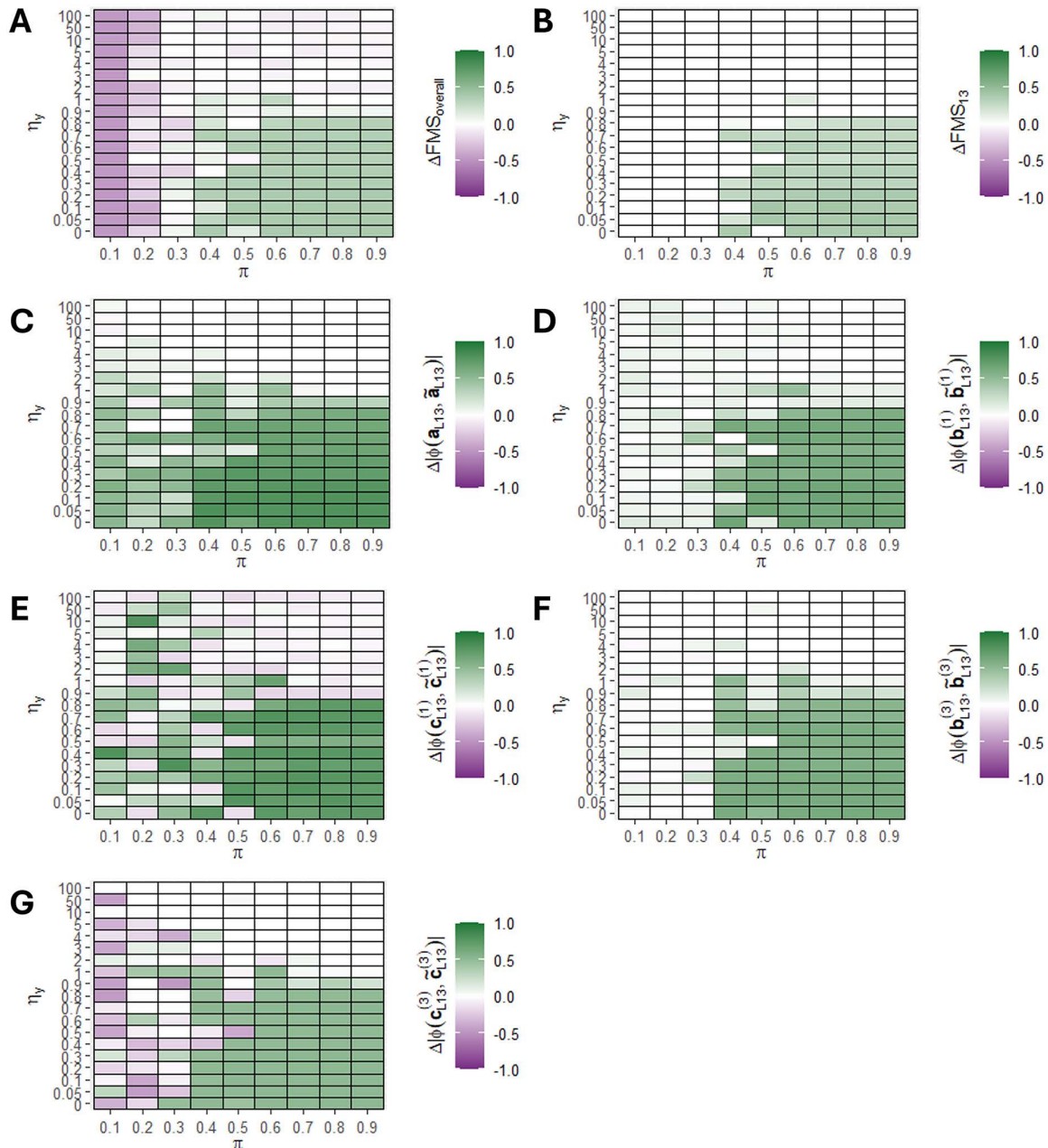

**Fig 6. The improvement in factor recovery by ACMTF-R under different amounts of noise on y.** One hundred randomly initialised eight-component ACMTF-R and ACMTF models were fitted for each combination noise on $\mathbf{y}$ ($\eta_y$) and tuning parameter ($\pi$), after which the lowest-loss models were compared. The improvement in factor recovery is computed by subtracting ACMTF performance from ACMTF-R performance in all panels. Improvement per setting for **(A)** overall factor recovery, **(B)** factor recovery of the hidden component, **(C)** shared subject mode recovery of the hidden component, **(D)** feature mode recovery of $\underline{\mathbf{X}}^{(1)}$ for the hidden component, **(E)** time mode recovery of $\underline{\mathbf{X}}^{(1)}$ for the hidden component, **(F)** feature mode recovery of $\underline{\mathbf{X}}^{(3)}$ for the hidden component, **(G)** time mode recovery of $\underline{\mathbf{X}}^{(3)}$ for the hidden component. Improvement compared to ACMTF is shown in green, while deterioration is shown in purple. A version of this figure without subtraction of ACMTF performance is shown in **Supplementary Figure 11 in** S1 File.

(ppBMI) as the outcome ($y$). To contextualise performance, we evaluated the results against ACMTF, which focuses only on the CLD structure, and N-way Partial Least Squares (NPLS) models per block [52], which focus only on prediction. We expected ACMTF-R to identify more strongly maternal ppBMI associated components compared to ACMTF.

A combined random initialisation and cross-validation procedure was used to select the number of components for ACMTF and ACMTF-R (**Methods**). The following tuning parameter values were investigated: $\pi = 0.75$, $\pi = 0.80$, $\pi = 0.85$, and $\pi = 0.90$. For each $\pi$ value, 10 randomly initialised models were fitted and a 10-fold cross-validation was performed across models with one to five components (**Supplementary Figure 12 in S1 File**). This analysis revealed that one component was optimal for all ACMTF-R settings. For ACMTF, a similar procedure indicated that two components were appropriate, each explaining a different source of variation (**Supplementary Figure 13 in S1 File**). For NPLS, cross-validation indicated that one component minimised RMSECV for the infant faecal and HM microbiome data, while two components were required for the HM metabolomics data (**Supplementary Figure 14 in S1 File**).

After selecting the number of components, 100 randomly initialised one-component ACMTF-R models were fitted for each $\pi$ value. The model with the lowest overall loss in each case was selected, and the variance explained was computed (**Table 1**). To contextualise performance, the ACMTF model explained 14.34%, 9.24%, and 7.03% of the variation in the infant faecal microbiome, HM microbiome, and HM metabolomics data blocks, respectively, while explaining 9.28% of the variation in $y$. Additionally, the NPLS models explained 5.78%, 5.41%, and 8.25% of the variation in the infant faecal microbiome, HM microbiome, and HM metabolomics datasets, respectively, while explaining 11.97%, 11.22%, and 35.95% of the variation in $y$.

The ACMTF-R model with $\pi = 0.90$ explained 2.08–8.31% of the variation in the independent data blocks, and 24.08% of the variation in $y$. At $\pi = 0.85$, this dropped to between 1.78–5.38% of the variation in the independent data blocks, and 62.06% in $y$. At $\pi = 0.80$ and $\pi = 0.75$, the models were most heavily tuned towards predicting $y$, yielding 1–2% of explained variation in the independent data blocks and capturing 90 + % of the variation in $y$ – a clear indication of overfitting.

MLR was performed on the subject mode loadings to test for associations with subject metadata, including maternal pre-pregnancy BMI, infant growth at six months, birth mode, maternal secretor (*Se*), and maternal Lewis (*Le*) blood group status (**Table 2**). Maternal *Se* and *Le* status determine the presence of specific fucosylated human milk oligosaccharides, which may influence the composition of the infant faecal microbiome.

The subject mode loadings of the NPLS models all contained at least one component that was associated with maternal pre-pregnancy BMI. While both components of the ACMTF model were associated with both ppBMI and maternal secretor status, the first component was most strongly associated with maternal secretor status and the second was most strongly associated with ppBMI. This was not the case for the ACMTF-R models with $\pi = 0.90$ and $\pi = 0.85$, which both

**Table 1. Variance explained per dataset for the fitted NPLS, ACMTF and ACMTF-R models.** $F$: the number of components.

| Dataset | Model | $F$ | Variation explained (%) | | | |
| --- | --- | --- | --- | --- | --- | --- |
| | | | Infant faecal microbiome | Human milk microbiome | Human milk metabolome | ppBMI ($y$) |
| Infant faecal microbiome | NPLS | 1 | 5.78 | – | – | 11.97 |
| HM microbiome | NPLS | 1 | – | 5.41 | – | 11.22 |
| HM metabolome | NPLS | 2 | – | – | 8.25 | 35.95 |
| All | ACMTF-R ($\pi = 0.75$) | 1 | 1.52 | 1.15 | 1.24 | 96.55 |
| | ACMTF-R ($\pi = 0.80$) | 1 | 2.45 | 1.91 | 1.41 | 89.97 |
| | ACMTF-R ($\pi = 0.85$) | 1 | 5.38 | 4.35 | 1.78 | 62.06 |
| | ACMTF-R ($\pi = 0.90$) | 1 | 8.31 | 6.69 | 2.08 | 24.08 |
| | ACMTF | 2 | 14.34 | 9.24 | 7.03 | 9.28 |

**Table 2. MLR results for the subject mode loadings of the created multi-way models.** ppBMI: maternal pre-pregnancy body mass index. *Se*: maternal secretor status. *Le*: maternal Lewis status. WHZ: infant weight-for-height z-score at 6 months. Subject mode loadings were normalised to norm 1 prior to MLR fitting for comparability of the results between the models. Values indicate MLR coefficients times one thousand. Stars indicate the original p-values from the MLR model (*: $p < 0.05$, **: $p < 0.01$, ***: $p < 0.001$), while hats indicate Benjamini-Hochberg corrected p-values (^: $q < 0.05$, ^^: $q < 0.01$, ^^^: $q < 0.001$). Coefficients with a significant p-value after correction are made bold. The correlation matrix of the metadata is reported in Supplementary Table 2 in **S1 File**.

| Dataset | Model | F | ppBMI | Birth mode | *Se* | *Le* | WHZ |
|---|---|---|---|---|---|---|---|
| Infant faecal microbiome | NPLS | 1 | **5.9***,^^^** | 26.2 | 18.7 | −67.4 | 1.5 |
| HM microbiome | NPLS | 1 | **5.3***,^^^** | 22.5 | 10.1 | −25.4 | −0.6 |
| HM metabolome | NPLS | 1 | **8.7***,^^^** | 11.1 | 19.7 | −78.3 | **−12.6*,^** |
| | | 2 | −0.2 | −12.8 | **−111***,^^^** | 43.0 | 7.0 |
| All | ACMTF-R ($\pi = 0.75$) | 1 | **15.0***,^^^** | 6.7 | 3.0* | −18.5 | −0.9 |
| | ACMTF-R ($\pi = 0.80$) | 1 | **−14.7***,^^^** | −10.8 | −5.3* | 31.7 | 1.4 |
| | ACMTF-R ($\pi = 0.85$) | 1 | **−12.5***,^^^** | −18.1 | −12.3* | 59.6 | 2.4 |
| | ACMTF-R ($\pi = 0.90$) | 1 | **−8.3***,^^^** | −21.5 | −21.8 | 79.4 | 2.8 |
| | ACMTF | 1 | **−2.9*,^** | −2.0 | **−74.8***,^^^** | 65.5 | 3.2 |
| | | 2 | **−4.4***,^^** | −40.1 | **−41.3*,^** | 72.3 | 1.0 |

contained subject mode loadings associated exclusively with ppBMI after correction. The subject mode loadings of the ACMTF-R models with lower values of $\pi$ were strongly associated with ppBMI due to overfitting.

Due to the extraction of a component exclusively related to maternal ppBMI, as well as the trade-off between explaining the independent data and predicting $y$, the ACMTF-R model using $\pi = 0.90$ was selected for further interpretation.

The $\Lambda$-matrix of the ACMTF-R model using $\pi = 0.90$ was examined to interpret the common, local, and distinct (CLD) structure of the component. In this model, $\lambda$ values of the only available component were equal to 0.34, 0.29, and 0.15, for the infant faecal microbiome, HM microbiome, and HM metabolomics, respectively. Based on the relative magnitudes of the $\lambda$ values, the component appears to be primarily local to the infant faecal and HM microbiome blocks, with a minor contribution of the HM metabolomics block. This suggests that ppBMI-associated variation was mainly present in the microbiomes of mother and infant.

In the infant faecal microbiome, genera associated with higher ppBMI included *Bifidobacterium bifidum*, *Escherichia coli, Staphylococcus epidermidis,* and *B. breve* (**Fig 7A**). Genera enriched in infants of low-ppBMI mothers included *Parabacteroides, Enterococcus* and *Streptococcus*. In the HM microbiome, higher ppBMI was associated with increased abundance of *Staphylococcus* and *Streptococcus* spp., while genera associated with lower ppBMI included various taxa such as *Clostridium sensu stricto*, *Collinsella*, and *Lactobacillus* (**Fig 7C**). In the HM metabolome, higher ppBMI was associated with increased levels of many unknown HMOs, caffeine, and LDFT (**Fig 7E**). Metabolites with increased levels in low-ppBMI mothers included several amino acids and related compounds such as 2-aminobutyrates, aspartate, isoleucine, and methionine.

Due to the pre-processing of the data, the largest absolute time loading in the ACMTF-R model indicates the time point at which inter-subject differences related to ppBMI are greatest (**Fig 7B**, **7D**, **7F**). Overall, the time loadings were relatively flat, suggesting that ppBMI-associated variation remained stable throughout the sampling period. In the HM microbiome, however, these differences peaked at day 30 and declined thereafter.

Taken together, ACMTF-R was able to identify variation that was strongly and exclusively associated with maternal ppBMI. Additionally, it revealed that this variation was primarily local to the infant faecal and HM microbiome, with a limited contribution from the HM metabolome. This highlights the value of incorporating supervision to isolate outcome-relevant signals in complex multi-omics data.

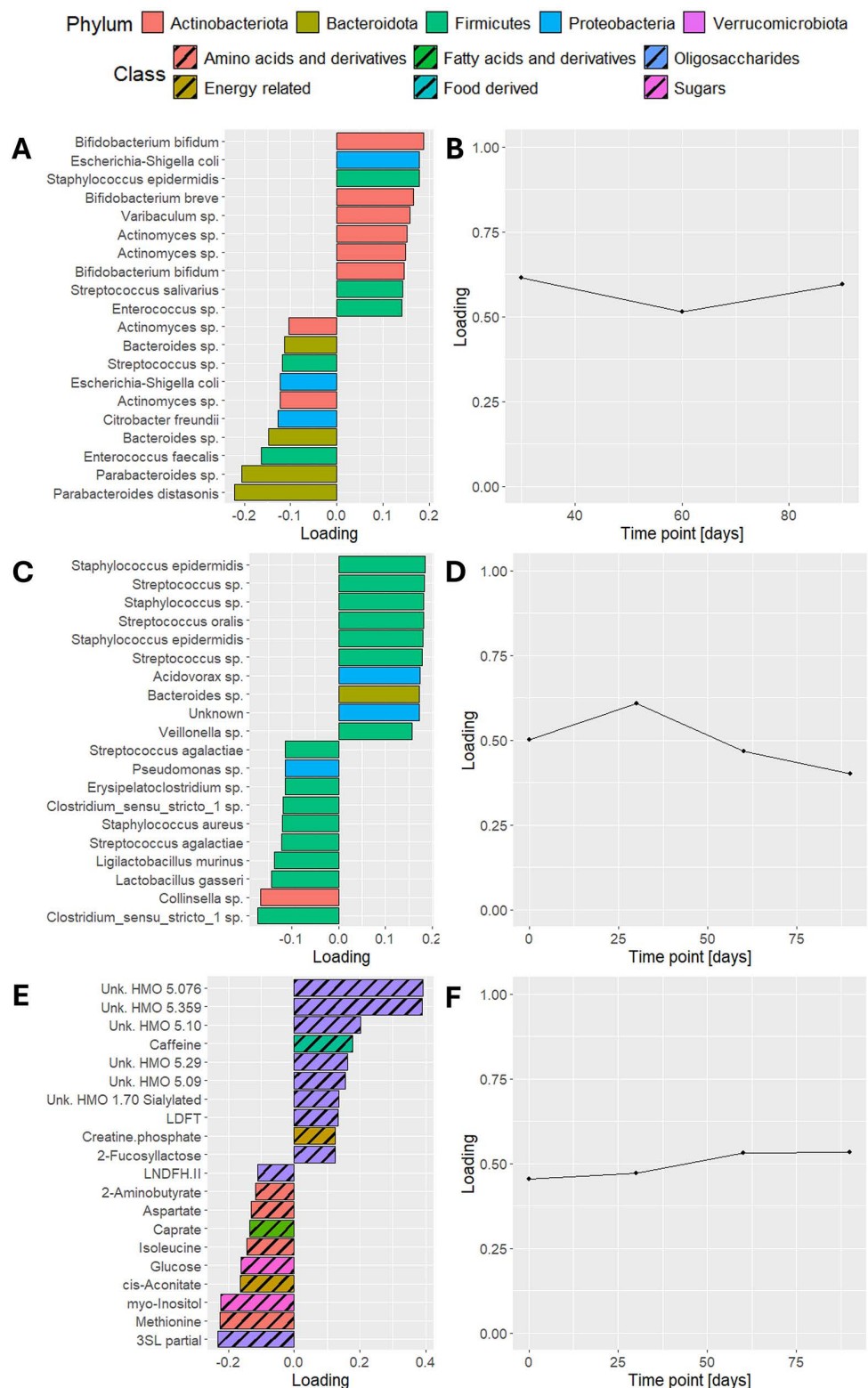

**Fig 7. Feature and time mode loadings of the selected ACMTF-R model.** Feature mode loadings of the selected ACMTF-R model using $\pi = 0.90$ describing **(A)** the infant faecal microbiome, **(C)** the HM microbiome, and **(E)** the HM metabolomics datasets. Time loadings of the same model are shown in the same order in **(B,D,F)**. Signs of the feature mode loadings were adjusted post-hoc such that positive values consistently corresponded to higher ppBMI and negative values to lower ppBMI.

## Discussion

In this study, we presented ACMTF-Regression (ACMTF-R) as an extension of Advanced Coupled Matrix and Tensor Factorization (ACMTF) that incorporates a regression term to model common, local and distinct (CLD) variation related to a dependent variable. Through simulations, we demonstrated its ability to capture a hidden component, its robustness to noise, and its flexibility in tuning between data exploration and predicting. Our application demonstrated how ACMTF-R finds novel relationships for the effect of maternal ppBMI on the HM metabolome, HM microbiome and infant faecal microbiome.

ACMTF-R extends the capabilities of ACMTF by allowing explicit modelling of outcome-associated variation, bridging the gap between exploratory data integration and supervised learning [10]. Compared to NPLS, which is designed for predictive modelling but lacks the ability to separate common, local, and distinct variation across multiple blocks, ACMTF-R provides a more nuanced view of multi-way data structures [19]. Unlike classical ACMTF, which only uncovers underlying data patterns, ACMTF-R allows for targeted analysis of biological variation of interest [37]. The ability to tune between the two makes it particularly useful for cases where a balance between exploration and prediction is necessary.

Our simulations revealed that ACMTF-R reliably recovers the underlying CLD-structure of multi-way datasets across a range of settings. We found that the tuning parameter ($\pi$) governs the trade-off between reconstructing the input data and predicting the dependent variable. While $\pi = 1$ corresponds to regular ACMTF, decreasing $\pi$ enhances the identification of factors associated with the dependent variable, even revealing weak structures that would otherwise remain undetected. However, tuning too far towards **y** ($\pi < 0.3$) leads to overfitting and an inaccurate reconstruction. The observation that intermediate values of $\pi$ (0.6–0.9) lead to recovery of the hidden local component is largely in agreement with PCovR and MCovR [24,25].

Furthermore, the simulations revealed that the hidden component needed to be sufficiently small ($\delta < 0.05$) for ACMTF to fail in its recovery. However, this finding is largely dependent on the relative sizes of the other components, which we have kept at Frobenius norm 1 in our simulations. These norms were intentionally controlled to showcase how supervision helps recover a small, outcome-related component to the reader. Since the true CLD structure of the data is not known to the model, this design is appropriate for addressing factor recovery under a well-specified ground truth. However, further research is needed to investigate the recovery of a hidden component using ACMTF and ACMTF-R in other scenarios.

Finally, the simulations showed that recovery of the hidden component using ACMTF-R was most accurate when a biologically realistic amount of noise was present in the data ($\eta_X \leq 0.5$, $\eta_y < 1$). The observation that neither ACMTF nor ACMTF-R can identify the hidden component in the noiseless case is surprising but has limited impact on real data applications. We speculate that noise may have a regularizing effect on flatter parts of the loss landscape, or that it may reduce factor collinearity in a phenomenon similar to ridge regression [60]. Due to non-convexity of the optimization problem, this is difficult to investigate and is left as future work. It is reassuring that ACMTF-R can reliably capture the **y**-associated subject mode loadings for most tested noise levels (**Supplementary Figure 10-11 in S1 File**).

Application to a longitudinal multi-omics dataset integrating human milk microbiome, human milk metabolome, and infant gut microbiome data further highlighted the utility of ACMTF-R by recovering an exclusively ppBMI-associated component [52]. In the infant faecal microbiome, elevated abundances of *Enterococcus* sp., *E. coli*, *Clostridium* sp., and *Staphylococcus epidermidis* in infants born to high-ppBMI mothers are consistent with prior findings linking maternal obesity to pro-inflammatory microbial profiles [61,62]. However, the association of elevated infant gut microbiome abundances of *Bifidobacterium breve* and *Bifidobacterium bifidum* with high maternal ppBMI is an unexpected result, since they are considered beneficial to early-life gut function [61–67]. It is possible that associations were confounded by reduced initiation or shorter duration of breastfeeding in overweight mothers, since cessation of breastfeeding strongly correlated with a decrease in bifidobacteria [63,67–70]. Future research should investigate the robustness of these associations using sensitivity analysis, or by using a cohort with homogenized breastfeeding initiation and duration.

Interestingly, the genera enriched in infants from low-ppBMI mothers represent a potential novel finding, as comparable reports are currently lacking, and warrant further investigation. In the HM microbiome, enrichment of *Streptococcus* and *Staphylococcus* spp. in high-ppBMI mothers is in agreement with previous research [71–73]. However, comparable data for taxa enriched in low-ppBMI mothers were not available for direct comparison.

In the HM metabolomics data, elevated levels of lactate and multiple unidentified HMOs were observed in high-ppBMI mothers. While the relationship between ppBMI and HMO concentrations has been investigated in several studies, the results remain inconsistent [74,75]. One study did report higher levels of a lactate derivative in the milk of obese mothers [74], suggesting a possible link to altered maternal energy metabolism. Conversely, the observation of increased concentrations of several amino acids in low-ppBMI mothers is supported by previous findings [74], as well as by recent evidence showing enrichment of amino acid biosynthesis pathways in this group [76].

Furthermore, the relatively flat time loadings across the blocks indicate that inter-subject differences remain approximately constant throughout the time series. The exception is the HM microbiome, where a transient increase is observed at day 30, potentially reflecting ppBMI effects on milk maturation, or dietary, or hormonal variation. Further research with more samples during the first month, ideally including information from dietary surveys and hormonal measurements, is needed to investigate this pattern in more detail. Taken together, these results show that ACMTF-R provides additional biological insights compared to ACMTF and NPLS, effectively disentangling common, local and distinct variation related to maternal ppBMI.

Finally, treating $y$ as an additional data block for ACMTF could offer a different approach to incorporating outcome variation into ACMTF-R. By embedding $y$ within the multi-way decomposition framework, rather than optimizing it separately, the model might be able to better capture latent structures linking $y$ to the input datasets. However, this approach would lose the predictive aspect of ACMTF-R. Future work should focus on the comparability between ACMTF-R and adding $y$ as a block to ACMTF.

## Limitations

While ACMTF-R demonstrates strong performance in identifying biologically meaningful patterns, several challenges remain. The choice of the tuning parameter $\pi$ is dataset-dependent, requiring careful selection through cross-validation or heuristic approaches. Additionally, in multi-omics applications, different data blocks may contain varying amounts of biologically relevant information. Allowing block-specific $\pi$ values could improve model flexibility, enabling certain datasets to contribute more strongly to the dependent variable while ensuring others remain focused on data structure. Future research should explore whether adaptive weighting strategies or block-specific hyperparameter tuning can further improve performance.

For ACMTF and ACMTF-R, the single guiding metric for selecting the number of components was often the stability of the decomposition through the factor match score (FMS) [15,59]. This paper has expanded upon this approach by formalising FMS for a cross-validation and random initialisation scheme, as well as developing the degeneracy score diagnostic to identify overfactorization. However, this approach is computationally intensive for the simulations. Future work should investigate if this approach could be accelerated through the Limited-memory Broyden-Fletcher-Goldfarb-Shanno [44–46] and AO-ADMM algorithms [22,23,77].

While Factor Match Score (FMS) has been a useful diagnostic for model selection, it suffers from the phenomenon of all blocks contributing to all components in ACMTF outlined above. Due to the way the structural model of ACMTF is implemented, $F$ trilinear components are fitted per data block, even when the rank of that block is lower than $F$ [37] This causes the creation of superfluous feature and time mode loading vectors that contribute to the model due to having non-zero $\lambda$ values. As a result, the FMS is negatively impacted by considering these components in the calculation. This problem may be avoided by more strongly penalising the $\Lambda$-matrix through replacement of the L1 penalty term by a smooth L0 penalty, as presented in Relaxed ACMTF (RACMTF) [78,79]. Future research should investigate if this approach yields more stable ACMTF and ACMTF-R models for multi-omics data.

The multi-omics application presented in this study highlights a critical shortcoming of ACMTF in the identification of common, local, and distinct (CLD) sources of variation by revealing that every data block contributes to all identified components. However, the relative magnitude of the contributing blocks captured by $\lambda$ still offers a useful interpretation. For example, a $\Lambda$-matrix threshold could be applied post-hoc to identify CLD-structures [37]. Another approach could be to pre-define the expected number of shared and distinct components, as suggested by Common and Discriminative subspace Non-negative Tensor Factorization (CDNTF) [80]. Future research should investigate if either approach or a combination of both might yield more interpretable ACMTF models for multi-omics data.

In this work we assume a single shared subject mode across data blocks, due to this setting being the most common in multi-omics data analysis. However, other applications may involve different, multiple, or only partially shared modes. Such settings have already been integrated into a well-posed and scalable implementation of CMTF using an Alternating Optimization (AO) Alternating Direction Method of Multipliers (ADMM) framework [22,23]. This approach is also capable of supporting other data distributions, such as Poisson and Negative Binomial. Ongoing work therefore targets the implementation of ACMTF-R using this framework to support other study designs.

## Conclusions

Our results demonstrate that ACMTF-R is a powerful framework for multi-way data integration that extends ACMTF by incorporating outcome-associated variation. Through simulations, we showed that ACMTF-R can recover a small hidden component, is robust to noise, and provides flexible tuning between data exploration and outcome prediction. Its application to multi-omics data highlights how ACMTF-R finds novel relationships for the effect of maternal ppBMI on the HM metabolome, HM microbiome and infant faecal microbiome. Ongoing work includes the implementation of ACMTF-R using an AO-ADMM framework to support other types of coupling.

## Supporting information

**S1 File. Supplementary Methods, Figures, and Tables.**
(DOCX)

**S2 File. Appendix 1: Derivation of the gradient function for ACMTF-R.**
(PDF)

## Acknowledgments

The authors wish to thank Daniël Rademaker (RU/UvA) for their useful discussions and suggestions, and Evrim Acar for her feedback regarding the optimization approach and convergence properties.

## Author contributions

**Conceptualization:** Geert Roelof van der Ploeg, Johan A. Westerhuis, Anna Heintz-Buschart, Fred T.G. White, Age K. Smilde.

**Formal analysis:** Geert Roelof van der Ploeg, Johan A. Westerhuis, Anna Heintz-Buschart.

**Funding acquisition:** Age K. Smilde.

**Methodology:** Geert Roelof van der Ploeg, Johan A. Westerhuis, Fred T.G. White, Age K. Smilde.

**Project administration:** Anna Heintz-Buschart.

**Software:** Geert Roelof van der Ploeg, Fred T.G. White, Rasmus Riemer Jakobsen.

**Supervision:** Johan A. Westerhuis, Anna Heintz-Buschart, Age K. Smilde.

**Visualization:** Geert Roelof van der Ploeg.

**Writing – original draft:** Geert Roelof van der Ploeg, Johan A. Westerhuis, Anna Heintz-Buschart, Fred T.G. White.

**Writing – review & editing:** Rasmus Riemer Jakobsen, Age K. Smilde.

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
