## [Decision Letter · Decision Letter 0]

17 Oct 2025

Dear Dr. Heintz-Buschart,

Thank you for submitting your manuscript to PLOS ONE. After careful consideration, we feel that it has merit but does not fully meet PLOS ONE’s publication criteria as it currently stands. Therefore, we invite you to submit a revised version of the manuscript that addresses the points raised during the review process.

**ACADEMIC EDITOR:** Thank you for submitting your manuscript to PLOS ONE. After evaluation, I am inviting a Major Revision.

We look forward to receiving your revised manuscript.

Kind regards,

Dr Sefki Kolozali

Academic Editor

PLOS ONE

Journal Requirements:

“GRvdP was funded by a grant from the University of Amsterdam, Research Priority

Area on Personal Microbiome Health.

FW was funded by a grant from the University of Amsterdam, Data Science Centre.”

Reviewers' comments:

Reviewer's Responses to Questions

**Comments to the Author**

1. Is the manuscript technically sound, and do the data support the conclusions?

Reviewer #1: Yes

Reviewer #2: Yes

2. Has the statistical analysis been performed appropriately and rigorously?

Reviewer #1: Yes

Reviewer #2: Yes

3. Have the authors made all data underlying the findings in their manuscript fully available?

Reviewer #1: Yes

Reviewer #2: Yes

4. Is the manuscript presented in an intelligible fashion and written in standard English?

Reviewer #1: No

Reviewer #2: Yes

Reviewer #1: Comments on the paper PONE-D-25-48000

Title: ACMTF-R: supervised multi-omics data integration uncovering shared and distinct outcome-associated variation

After my review, I think this article is worth publishing subject to the following queries.

Comment 1: There are still a few grammatical slips (indefinite vs. definite articles, singular/plural mismatch, missing hyphens) throughout the manuscript. Please proof-read once more.

Comment 2: The Abstract clearly states the methodological gap but does not mention the actual biomedical context that motivates the work. Add one or two sentences that describe why linking maternal BMI to infant/milk multi-omics is an urgent question.

Comment 3: In the Introduction, insert a short paragraph that explicitly lists the three novel points of the paper: (i) regression extension of ACMTF, (ii) tuning parameter that balances exploration vs. prediction, (iii) unified CLD extraction in multi-block tensors. Relate these points to the most recent literature (including the three references below).

Comment 4: The Simulation section should compare ACMTF-R with at least one other supervised multi-block method (e.g., N-way PLS or SGCCA) in terms of component recovery and RMSE, and comment on computational cost.

Comment 5: The Conclusion should add one sentence on the limitation of the single shared-mode assumption and indicate whether a multi-mode extension is underway.

Comment 6: References should be updated to include the following three papers that are directly relevant to age-structured and fractional-order modelling of behavioural addictions and epidemic strains:

- Fractional-order modeling and optimal control of a new online game addiction model based on real data. Commun Nonlinear Sci. 2023;121(8):107221.

- Modeling the competitive transmission of the Omicron strain and Delta strain of COVID-19. J Math Anal Appl. 2023;526(2):127283.

Reviewer #2: INTRODUCTION

Weakness

1. Some phrases (e.g., on ACMTF's emergence and the gap between exploration/prediction) overlap closely with the abstract, which could be streamlined to avoid redundancy.

2. While citations are appropriate (e.g., 1-13 for background), they cluster early; adding a few more recent references (post-2020) on multi-omics integration or tensor methods could strengthen timeliness, especially given the field's rapid evolution.

METHODS

Weakness

1. The methods assume a shared first mode (subjects) across all blocks and y, restricting applicability to datasets with mismatched modes or higher-order tensors. While noted, no discussion of extensions (e.g., to non-shared modes) is provided, potentially limiting generalizability.

2. The tuning parameter π requires manual selection via simulations, with risks of overfitting at low values (e.g., π<0.3). Random initializations (100 per model) address local minima but increase computational burden; no automated selection (e.g., via BIC or elbow method) is proposed beyond cross-validation for component number.

3. Simulations fix most components at norm 1, with only one varying (δ), and use Gaussian noise, which may not capture real-world complexities like correlated noise or non-Gaussian distributions. Recovery issues in low-noise cases (e.g., η_X=0) are observed but not deeply analyzed (e.g., via optimization landscape). The fixed CLD structure (2 common, 3 local, 3 distinct) could be expanded to test robustness to varying structures.

4. While TCC and FMS are used effectively, their sensitivity (e.g., TCC to sign flips, addressed via absolute value) is mentioned briefly. The Hungarian algorithm for matching is appropriate but assumes known CLD types; real applications might require additional validation (e.g., against biological priors).

5. The application preprocessing (e.g., arbitrary sparsity thresholds of 75-85%, pseudo-count=1) is justified but dataset-specific; broader guidance on handling compositional data or missing values in multi-omics would strengthen the section. The intersection of subjects (reducing to 158) is necessary but could introduce bias if dropouts are non-random.

RESULTS AND DISCUSSION

This section can further be improved.

Four simulations comprehensively test ACMTF-R's performance under varying conditions (hidden component size δ, noise on X η_X, noise on y η_y). Results clearly show improved recovery (e.g., FMS, TCC >0.95) at intermediate π (0.6-0.9), robustness to biological noise (30-50%), and limits (e.g., fails at δ<0.03 or no noise). Figures (3-6) and supplements effectively visualize improvements over ACMTF.

1. Some results (e.g., higher Bifidobacterium in high-ppBMI) contradict literature without deep explanation or validation. Discussion acknowledges but could strengthen with hypotheses or sensitivity analyses.

2. Low π leads to overfitting, but results don't quantify prediction error (e.g., RMSE on hold-out set) beyond variance explained. Application focuses on π=0.9, but comparative predictive performance vs. NPLS is limited.

3. Fixed structures (e.g., 8 components, norm 1 except hidden) may not generalize; no-noise failure is noted but unexplained (e.g., no optimization diagnostics). Lacks sensitivity to varying CLD patterns or higher ranks.

4. Single dataset; CLD interpretation relies on relative λ, but all blocks contribute (issue noted in discussion). Time loadings "flat" but peaked at day 30 in HM microbiome—could explore more.

5. Dense figures; supplements referenced heavily. No direct comparison of recovered vs. true in application (as in simulations).

I would have liked to see a more detailed discussion regarding the results presented in this study and if possible the implications of the results. In other words, the results are not well discussed.

The authors should carefully correct all typos in the current version of the work

**Do you want your identity to be public for this peer review?** For information about this choice, including consent withdrawal, please see our Privacy Policy

Reviewer #1: No

Reviewer #2: **Yes: ** Kwame Assa-Agyei

---

## [Author Response · Author response to Decision Letter 1]

27 Nov 2025

REVIEWER 1

Reviewer #1: Comments on the paper PONE-D-25-48000

Title: ACMTF-R: supervised multi-omics data integration uncovering shared and distinct outcome-associated variation

After my review, I think this article is worth publishing subject to the following queries.

Comment 1: There are still a few grammatical slips (indefinite vs. definite articles, singular/plural mismatch, missing hyphens) throughout the manuscript. Please proof-read once more.

We thank the reviewer for this careful and constructive comment. We have thoroughly re-read the manuscript and corrected remaining issues related to article use, singular/plural agreement, and similar grammatical inconsistencies. These changes are numerous and therefore not individually highlighted, but nearly every paragraph has been polished accordingly.

Comment 2: The Abstract clearly states the methodological gap but does not mention the actual biomedical context that motivates the work. Add one or two sentences that describe why linking maternal BMI to infant/milk multi-omics is an urgent question.

Thank you for this remark. The development of ACMTF-R was motivated specifically by this application. Emerging evidence suggests that inter-generational transfer of maternal obesity may affect multiple omics layers, highlighting the need to identify outcome-associated variation [1]. We have adapted the Abstract accordingly (lines 36-40).

References:

Voerman, E. et al. Maternal body mass index, gestational weight gain, and the risk of overweight and obesity across childhood: An individual participant data meta-analysis. PLOS Med. 16, e1002744 (2019).

Comment 3: In the Introduction, insert a short paragraph that explicitly lists the three novel points of the paper: (i) regression extension of ACMTF, (ii) tuning parameter that balances exploration vs. prediction, (iii) unified CLD extraction in multi-block tensors. Relate these points to the most recent literature (including the three references below).

We thank the reviewer for this constructive suggestion. The novelty of ACMTF-R is the combination of four existing principles: (i) the separation of joint and individual structures, (ii) relating latent variables to an outcome, (iii) performing a coupled decomposition for any combination of matrices and tensors, and (iv) a tuning parameter that balances data reconstruction and prediction. In connection with comment #2 on the Introduction section by Reviewer 2, we have therefore revised the Introduction (lines 80-86) to more clearly indicate the contribution as well as the relevant recent literature that ACMTF-R is built upon. Please refer to our answer to comment #6 regarding the suggested references.

Comment 4: The Simulation section should compare ACMTF-R with at least one other supervised multi-block method (e.g., N-way PLS or SGCCA) in terms of component recovery and RMSE, and comment on computational cost.

We thank the reviewer for this comment. However, N-way PLS can only be applied block-wise and (s)GCCA is a matrix decomposition method that requires a user-specified design matrix linking the blocks in a specific way. In contrast, ACMTF-R is a joint matrix-tensor decomposition framework that enforces strict coupling of the shared subject mode across blocks, providing a unified latent space and enforces an explicit CLD structure under the reconstruction + regression objective. Consequently, a component-level recovery comparison would be methodologically uneven, and we therefore restrict the simulations to compare only ACMTF and ACMTF-R to highlight finding the hidden component. The additional computational cost of the regression step for y in ACMTF-R is negligible compared to ACMTF due to being closed-form. We have adapted lines 243-247 in Methods to elaborate on this aspect.

Comment 5: The Conclusion should add one sentence on the limitation of the single shared-mode assumption and indicate whether a multi-mode extension is underway.

We thank the reviewer for this suggestion. Connected to methods comment #1 by reviewer 2, we added a Limitations section to the manuscript to elaborate on these (and other) aspects of the study.

We agree that the current implementation is limited to only sharing the first mode. This is because our primary motivation for developing ACMTF-R contained a shared first mode, and because this setting is the most common in multi-omics study designs. Other shared modes, multiple shared modes, or partially shared modes are also possible in some circumstances, and the algorithm would have to be adapted to accommodate such settings. Recent work adapted CMTF to such settings using an Alternating Optimization (AO) Alternating Direction Method of Multipliers (ADMM) framework [8,9]. Ongoing work therefore targets the implementation of ACMTF-R into this framework. We have added the paragraph in lines 808-814 of Limitations to elaborate on this aspect. In addition, we have added lines 821-822 to Conclusions.

Comment 6: References should be updated to include the following three papers that are directly relevant to age-structured and fractional-order modelling of behavioural addictions and epidemic strains:

- Fractional-order modeling and optimal control of a new online game addiction model based on real data. Commun Nonlinear Sci. 2023;121(8):107221.

- Modeling the competitive transmission of the Omicron strain and Delta strain of COVID-19. J Math Anal Appl. 2023;526(2):127283.

We thank the reviewer for their suggestion. However, the cited papers concern fractional-order and age-structured compartmental differential equation models for behavioral addiction and epidemic dynamics, which are not pertinent to the tensor-based data fusion framework developed in this work. We therefore believe that these references may have been included in error or intended for a different manuscript. Should the reviewer see a specific conceptual link between these studies and the present work, we would be happy to consider further clarification in a future revision.

REVIEWER 2

Abstract:

The abstract is well organized. More importantly, it makes it easy to effectively distinguish or identify the major components of an abstract, which include the background, aims/goals, method, results, and conclusion.

INTRODUCTION

Weakness

Some phrases (e.g., on ACMTF's emergence and the gap between exploration/prediction) overlap closely with the abstract, which could be streamlined to avoid redundancy.

Thank you for this important suggestion. We have adapted lines 21-25 of the Abstract to avoid this redundancy. In connection with comment #3 by Reviewer 1, lines 80-86 in the Introduction were adapted to avoid some similar phrasing as well.

While citations are appropriate (e.g., 1-13 for background), they cluster early; adding a few more recent references (post-2020) on multi-omics integration or tensor methods could strengthen timeliness, especially given the field's rapid evolution.

We thank the reviewer for this comment. In connection with comment #3 by Reviewer 1, we have adapted lines 80-86 of the Introduction section to more clearly position ACMTF-R on the intersection of four strands of prior work: (i) the separation of joint and individual structures, (ii) relating latent variables to an outcome, (iii) performing a coupled decomposition for any combination of matrices and tensors, and (iv) a tuning parameter that balances data reconstruction and prediction. For each of these, relevant recent references have been added.

METHODS

Weakness

The methods assume a shared first mode (subjects) across all blocks and y, restricting applicability to datasets with mismatched modes or higher-order tensors. While noted, no discussion of extensions (e.g., to non-shared modes) is provided, potentially limiting generalizability.

Thank you for this suggestion. In connection with comment #5 by Reviewer 1, we have revised the Discussion section to now include Limitations.

We agree that the current implementation is limited to only sharing the first mode. This is because our primary motivation for developing ACMTF-R contained a shared first mode, and because this setting is the most common in multi-omics study designs. Other shared modes, multiple shared modes, or partially shared modes are also possible in some circumstances, and the algorithm would have to be adapted to accommodate such settings. Recent work adapted CMTF to such settings using an Alternating Optimization (AO) Alternating Direction Method of Multipliers (ADMM) framework [2,3]. Ongoing work therefore targets the implementation of ACMTF-R into this framework. We have added the paragraph in lines 808-814 of Limitations to elaborate on this aspect. In addition, we have added lines 821-822 to Conclusions.

References:

Schenker C, Cohen JE, Acar E. An Optimization Framework for Regularized Linearly Coupled Matrix-Tensor Factorization. In: 2020 28th European Signal Processing Conference (EUSIPCO) [Internet]. Amsterdam, Netherlands: IEEE; 2021 [cited 2025 July 24]. p. 985–9. Available from: https://ieeexplore.ieee.org/document/9287459/

Schenker C, Cohen JE, Acar E. A Flexible Optimization Framework for Regularized Matrix-Tensor Factorizations With Linear Couplings. IEEE J Sel Top Signal Process. 2021 Apr;15(3):506–21.

The tuning parameter π requires manual selection via simulations, with risks of overfitting at low values (e.g., π<0.3). Random initializations (100 per model) address local minima but increase computational burden; no automated selection (e.g., via BIC or elbow method) is proposed beyond cross-validation for component number.

Thank you for emphasizing this aspect of the study. In real multi-omics applications, a single optimal number of components does not exist, since this depends on data idiosyncrasies (block scale, noise, sharing of modes, outcome) as well as biological interpretability of the model. Consequently, information criteria such as BIC and heuristics such as the elbow methods are difficult to apply meaningfully. We instead recommend an analyst-guided triangulation strategy for ACMTF-R that balances optimal model stability (FMS), predictive power (RMSECV), fit, and biological interpretability of the recovered components. This procedure mitigates overfitting and prioritises solutions that are stable and interpretable. We have adapted lines 407-412 and lines 457-458 in Methods to better clarify these aspects.

Simulations fix most components at norm 1, with only one varying (δ), and use Gaussian noise, which may not capture real-world complexities like correlated noise or non-Gaussian distributions. Recovery issues in low-noise cases (e.g., η_X=0) are observed but not deeply analyzed (e.g., via optimization landscape). The fixed CLD structure (2 common, 3 local, 3 distinct) could be expanded to test robustness to varying structures.

We agree that real data can deviate from the simple Gaussian noise setting and that the CLD structure may vary. Our simulation design was intentionally controlled to isolate the question of interest: how supervision helps recover a small, outcome-related component. To achieve this, we fixed the other components to unit norm to make the comparisons across π and noise levels as transparent as possible for the reader. Since the true CLD structure of the data is not known to the model, this design is appropriate for addressing factor recovery under a well-specified ground truth. We have added these aspects to the Discussion section (lines 720-723).

While we used Gaussian noise as a baseline, extending ACMTF-R to non-Gaussian and/or correlated noise (e.g., Poisson or Negative Binomial count blocks) requires alternative loss functions and constraints (e.g., KL-divergence) and possibly a mixed-loss AO-ADMM solver in practice. This is beyond the scope of the present study and is noted in Limitations (lines 808-814).

We acknowledge and agree that the low-noise result has only received a shallow interpretation in the Discussion section. This is due to the ACMTF-R optimization function being non-convex, making the effect of noise on the landscape unclear. While we suspect that noise has a regularizing effect on flatter parts of the landscape, this is difficult to prove in practice (and beyond the scope of this study). We also note that strictly noiseless data is rare in practice, so this edge case has limited impact on real data applications. We have revised the Discussion paragraph on noise (lines 727-733) to better clarify these aspects.

While TCC and FMS are used effectively, their sensitivity (e.g., TCC to sign flips, addressed via absolute value) is mentioned briefly. The Hungarian algorithm for matching is appropriate but assumes known CLD types; real applications might require additional validation (e.g., against biological priors).

Thank you for this helpful suggestion. We use absolute TCC to remove sign indeterminacy and compute FMS after matching using the Hungarian algorithm. In the simulations, the matching uses the factors as well as the CLD structure (since the ground truth is known). In the application, only the factors themselves are used. This is elaborated upon in the Supplementary Methods. We adapted lines 364-367, 424-426, and 438-440 to emphasize this difference in approach.

As mentioned in your comment #2, we agree that real applications should include biological information as part of the triangulation strategy to determine the correct number of factors. In the present study, this was done in close collaboration with our collaborators from the University of Copenhagen. Additionally, we quantify the biological effect modelled by each factor via associations with subject-level covariates (MLR).

The application preprocessing (e.g., arbitrary sparsity thresholds of 75-85%, pseudo-count=1) is justified but dataset-specific; broader guidance on handling compositional data or missing values in multi-omics would strengthen the section. The intersection of subjects (reducing to 158) is necessary but could introduce bias if dropouts are non-random.

We agree with the reviewer that these aspects of data preprocessing are critical. However, the focus of this work has been the methodology. Preprocessing of the Jakobsen2025 dataset is outlined in the original paper [4], and we elaborate in detail on handling compositional data in a separate paper [5]. We have adapted lines 376-378 in Methods to more clearly point towards these resources.

Your comment on taking the intersection of subjects across blocks is valid and important. In essence, this is a result of hard coupling the subject modes of the blocks. As we mention in our answer to your comment #1 and #3, a more flexible coupling approach using AO-ADMM was recently presented that would allow us to keep all subjects [6,7]. Ongoing work includes the implementation of ACMTF-R into this framework. These aspects have been added to the Limitations section (lines 808-814).

References:

Jakobsen R, Ploeg GR, Sundekilde U, Astono J, Poulsen K, Fuglsang J, et al. Supervised Modelling of Longitudinal Human Milk and Infant Gut Microbiome Reveal Maternal Pre-Pregnancy BMI and Early Life Growth Interactions [Internet]. Research Square; 2025 [cited 2025 Apr 6]. Available from: https://www.researchsquare.com/article/rs-6244750/v1

van der Ploeg GR, Westerhuis JA, Heintz-Buschart A, Smilde AK. parafac4microbiome: Exploratory analysis of longitudinal microbiome data using Parallel Factor Analysis [Internet]. 2024 [cited 2024 July 8]. Available from: http://biorxiv.org/lookup/doi/10.1101/2024.05.02.592191

Schenker C, Cohen JE, Acar E. An Optimization Framework for Regularized Linearly Coupled Matrix-Tensor Factorization. In: 2020 28th European Signal Processing Conference (EUSIPCO) [Internet]. Amsterdam, Netherlands: IEEE; 2021 [cited 2025 July 24]. p. 985–9. Available from: https://ieeexplore.ieee.org/document/9287459/

Schenker C, Cohen JE, Acar E. A Flexible Optimization Framework for Regularized Matrix-Tensor Factorizations With Linear Couplings. IEEE J Sel Top Signal Process. 2021 Apr;15(3):506–21.

---------

---

## [Decision Letter · Decision Letter 1]

9 Dec 2025

ACMTF-R: supervised multi-omics data integration uncovering shared and distinct outcome-associated variation

PONE-D-25-48000R1

Dear Dr. Heintz-Buschart,

We’re pleased to inform you that your manuscript has been judged scientifically suitable for publication and will be formally accepted for publication once it meets all outstanding technical requirements.

Kind regards,

Sefki Kolozali

Academic Editor

PLOS One

Additional Editor Comments (optional):

Reviewers' comments:

Reviewer's Responses to Questions

**Comments to the Author**

Reviewer #1: All comments have been addressed

2. Is the manuscript technically sound, and do the data support the conclusions?

Reviewer #1: Yes

3. Has the statistical analysis been performed appropriately and rigorously?

Reviewer #1: Yes

4. Have the authors made all data underlying the findings in their manuscript fully available?

Reviewer #1: Yes

5. Is the manuscript presented in an intelligible fashion and written in standard English?

Reviewer #1: Yes

Reviewer #1: (No Response)

**Do you want your identity to be public for this peer review?** For information about this choice, including consent withdrawal, please see our Privacy Policy

Reviewer #1: No

---

## [Editor Report · Acceptance letter]

PONE-D-25-48000R1

PLOS One

Dear Dr. Heintz-Buschart,

I'm pleased to inform you that your manuscript has been deemed suitable for publication in PLOS One. Congratulations! Your manuscript is now being handed over to our production team.

Kind regards,

on behalf of

Dr. Sefki Kolozali

Academic Editor

PLOS One